# Differentiable Cluster Discovery in Temporal Graphs

**Md Hafizur Rahman**                                            *hafizur.rahman@mail.utoronto.ca*
*Department of Mechanical and Industrial Engineering*
*University of Toronto*

**Chi-Guhn Lee**                                                  *chiguhn.lee@utoronto.ca*
*Department of Mechanical and Industrial Engineering*
*University of Toronto*

**Reviewed on OpenReview:** *https://openreview.net/forum?id=1caZVb6zL7*

## Abstract

Existing temporal graph clustering methods suffer from poor optimization dynamics due to reliance on heuristically initialized cluster assignment distribution without considering the dynamic nature of the evolving graph. The target cluster assignment distribution often conflicts with evolving temporal representations, leading to oscillatory gradients and unstable convergence. Motivated by the need for differentiable and adaptive clustering in dynamic settings, we propose TGRAIL (**T**emporal **Gr**aph **A**lignment and **I**ndex **L**earning), a novel end-to-end framework for temporal graph clustering based on Gumbel–Softmax sampling. TGRAIL enables discrete cluster assignments while maintaining the gradient flow. To ensure stable training, we formulate the clustering objective as an expectation over Monte Carlo samples and show that this estimator is both unbiased and variance-reduced. Furthermore, we incorporate a temporal consistency loss to preserve the order of interactions across time. Extensive experiments on six real-world temporal graph datasets demonstrate that our approach consistently outperforms state-of-the-art baselines, achieving higher clustering accuracy and robustness. Our results validate the effectiveness of jointly optimizing temporal dynamics and discrete cluster assignments in evolving graphs.

## 1 Introduction

Graphs are fundamental tools for modeling relationships and interactions in complex systems, spanning domains such as social networks, biological networks, communication systems, and financial markets (Hamilton et al., 2017; Ying et al., 2019; Sun et al., 2020; Wang et al., 2022; Zheng et al., 2025). A central task in graph analysis is clustering, which aims to group nodes into communities based on structural or semantic similarity. Traditional graph clustering methods operate on static graphs, where the topology and node attributes remain fixed. These methods, including spectral clustering and modularity-based approaches (Bianchi et al., 2020; You et al., 2021; Tsitsulin et al., 2023; Liu et al., 2025a), have been widely adopted due to their theoretical foundations and interpretability.

Compared to this, deep clustering methods integrate representation learning with clustering objectives. For instance, Deep Embedded Clustering (DEC) (Xie et al., 2016) combines autoencoder-based embeddings with Kullback–Leibler (KL) divergence-based soft assignments. Extensions such as Improved DEC (Guo et al., 2017) and Structural Deep Clustering Networks (SDCN) (Bo et al., 2020) incorporate reconstruction losses or graph neural networks to better leverage node features and topology. Despite their success, these methods are fundamentally static: they assume access to a complete adjacency matrix and cannot model temporal dependencies. Consequently, they are unable to capture the evolving nature of communities or adapt to dynamic patterns of interaction.

Recently, temporal graph clustering has emerged and gained attention to address these limitations. A temporal graph captures the temporal dimension through a sequence of time-stamped events. Instead of

modeling edges as static relations, temporal graphs represent interactions as sequences, allowing finer-grained analysis of how relationships form, persist, and dissolve over time. This richer representation enables new opportunities, such as tracking evolving communities, detecting temporal anomalies, and forecasting future events (Cini et al., 2025; Postuvan et al., 2024; Liu et al., 2024).

Several approaches have been proposed to model temporal graphs. Time-aware graph neural networks (TGNNs), such as TGAT (Xu et al., 2020), TGN (Rossi et al., 2020), and HTNE (Zuo et al., 2018), introduce temporal attention, memory, or Hawkes processes to encode evolving features. However, these methods typically decouple representation learning from clustering, requiring a post hoc clustering step. This two-stage design can be suboptimal, as the learned representations may not align well with the clustering objective, and errors from the first stage propagate without correction. Moreover, the clustering step is non-differentiable, preventing end-to-end training.

Recent methods attempt to address this limitation by integrating clustering within the training loop. For example, TGC (Liu et al., 2024) incorporates a clustering loss into the temporal graph encoder using soft assignments derived from a Student's $t$-distribution. This approach enables joint optimization of embeddings and cluster centroids. While the target distribution is expressed as time dependent in their approach, its reliance on fixed node embeddings results in a distribution that does not evolve over time which fails to adapt temporally consistent cluster assignment. Additionally, the $t$-distribution has several drawbacks in dynamic settings: it assumes a fixed degree-of-freedom parameter, is sensitive to initialization, and tends to overemphasize outliers due to its heavy-tailed nature (Linderman & Steinerberger, 2019). Figure 1 we provide a visual depiction of temporal cluster dynamics in evolving graphs.

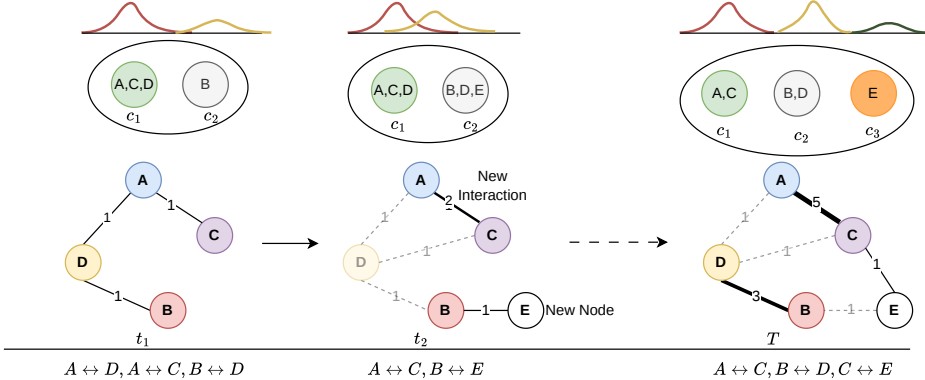

Figure 1: Temporal evolution of cluster assignments in dynamic graphs. Nodes may shift clusters due to new interactions, inactive nodes, or structural changes over time.. At $t_1$, nodes $A, C, D$ form cluster $C_1$, and $B$ belongs to $C_2$. A new node $E$ appears at $t_2$, reshaping interactions and leading to reassignment of $B, D, E$ to $C_2$. By time $T$, node $E$ forms a separate cluster $C_3$.

To address the aforementioned limitations, we propose a novel, differentiable framework for temporal graph clustering. We formulate the cluster assignment process as stochastic sampling from a Gumbel-Softmax distribution, which enables discrete assignments to be learned through gradient-based optimization. We summarize our contributions as follows-

1. **A differentiable framework for temporal graph clustering.** We propose TGRAIL, a method that *jointly* learns node representations and discrete cluster assignments in dynamic graphs via a Monte Carlo Gumbel Softmax re-parameterization. This removes the need for *post-hoc* process or t-distribution soft assignments, enabling end-to-end training thus aligns cluster assignment with temporal node embeddings.

2. **Unbiased, low-variance gradient estimation with theoretical guarantees.** We derive a tight variance bound for the Gumbel estimator and prove a non-asymptotic SGD convergence theorem under

standard Lipschitz and bounded-step assumptions. Our analysis clarifies why discrete assignments remain stable throughout training.

3. **A unified temporal–clustering loss that scales linearly in interactions.** By coupling a temporal contrastive objective with the discrete clustering loss, we keep complexity at $\mathcal{O}(|E|)$ rather than $\mathcal{O}(N^2)$, making TGRAIL practical for long, sparse interaction streams.

4. **Extensive empirical validation on six evolving-graph datasets.** TGRAIL outperforms ten SOTA baselines by 3–5% macro-$F_1$ on sparse datasets (PATENT, DBLP) and matches or exceeds the best methods on dense or highly non-stationary graphs.

## 2 Related Work

**Graph Clustering via Neural Networks.** Initial deep learning approaches to graph clustering leveraged MLP-based autoencoders to extract latent node embeddings from the graph structure. GraphEncoder (Tian et al., 2014) and DNGR (Cao et al., 2016) encoded proximity between nodes using sparse autoencoders and random walk-based techniques, followed by $k$-means clustering. These early methods demonstrated that deep representations could improve clustering performance but struggled to integrate node attribute information. The introduction of graph convolutional networks enabled models to jointly encode structural and attribute information. Kipf and Welling's VGAE (Kipf & Welling, 2016) and Wang et al.'s MGAE (Wang et al., 2017) used graph encoders to produce informative latent spaces for downstream clustering. These works laid the groundwork for reconstructive methods (Wang et al., 2019; Park et al., 2019; Bandara et al., 2024; Yu et al., 2025) where reconstruction of adjacency or feature matrices acted as the self-supervised objective. Similarly, adversarial mechanisms were introduced to regularize latent spaces and improve representation robustness. ARGA (Pan et al., 2018) employed a discriminator to align latent embeddings with a Gaussian prior, while CommunityGAN (Jia et al., 2019) generated synthetic samples for structure-preserving embedding. Though effective in reducing overfitting and capturing community semantics, these methods separate clustering from representation learning, which introduces unstable cluster assignments.

**Clustering-Oriented Architectures and Fusion Models.** On the other hand, several methods sought to unify representation learning with clustering objectives. DAEGC (Wang et al., 2019) proposed attention-based graph encoders guided by clustering alignment loss. GALA (Park et al., 2019) enhanced encoder expressiveness via Laplacian sharpening. Models like SDCN (Bo et al., 2020) and DFCN (Tu et al., 2021) integrated attribute and structure views using novel fusion strategies, demonstrating that explicit clustering supervision during representation learning improved cluster separation and compactness. As graph sizes increased, scalability became a central concern (Xu et al., 2025). S3GC (Devvrit et al., 2022) performed scalable contrastive learning using batch-wise subgraph sampling and post-hoc $k$-means clustering. Dink-Net (Liu et al., 2023) unified contrastive representation learning and clustering optimization via differentiable dilation and shrinkage losses, enabling end-to-end training on graphs with over 100M nodes.

**Dynamic/Temporal Graph Clustering.** Temporal graph clustering extends conventional graph clustering to dynamic scenarios where node interactions evolve over time. Liu et al. (Liu et al., 2024) propose a general framework called Temporal Graph Clustering (TGC). This framework integrates temporal representation learning with clustering objectives tailored for interaction-sequence data. MVTGC (Liu et al., 2025b), CGC (Park et al., 2022) utilize multiview and contrastive objectives between graph snapshots to capture evolving community structures. These models address the temporal nature of clustering, which static methods cannot handle effectively.

Despite these advances, GNN-based temporal graph clustering approaches model $t$-distribution as a cluster assignment distribution (Bo et al., 2020; Liu et al., 2024; 2025a), which may be suboptimal in dynamic settings due to its heavy tails that amplify the influence of transient or noisy nodes. In contrast to these approaches, we learn the cluster assignment using Gumbel Softmax in an end-to-end manner, aligning temporal evolution and cluster assignment.

## 3 Temporal Graph Clustering

### 3.1 Problem Definition

As stated in the previous section, temporal graphs capture not a fixed structure but an evolving stream of interactions. In such dynamic networks, whether social platforms, citation graphs, or sensor grids where nodes can emerge, disappear, or reconfigure their connections over time. This evolution manifests as fluctuations in node activity, shifting neighborhood contexts, and changing roles, all of which influence the cluster membership of each node at every timestamp. To capture this temporal evolution, we consider the network as a continuous sequence of graphs where the topology is a chronological stream of temporal events Rossi et al. (2020); Chmura et al. (2025). Let the sequence be $\{G^{(1)}, G^{(2)}, \ldots, G^{(T)}\}$, where each timestamp $G^{(t)} = (\mathcal{V}^{(t)}, \mathcal{E}^{(t)})$ represents the network's state at time $t$ where $\mathcal{V}^{(t)}$ denotes the set of active nodes, and $\mathcal{E}^{(t)} \subseteq \mathcal{V}^{(t)} \times \mathcal{V}^{(t)}$ defines their pairwise interactions. We can define the problem of temporal graph clustering as follows. For notation clarity, we denote matrices in bold capital letters, vectors in bold small letters, and scalars in non-bold letters.

**Problem 3.1** (Continuous-Time Temporal Graph Clustering)**.** Given a temporal graph $\mathcal{G} = (\mathcal{V}, \mathcal{E}, \mathcal{T})$ and time-dependent node features $\mathbf{X}^{(t)} \in \mathbb{R}^{N \times D}$ and adjacency matrix $\mathbf{A}^{(t)} \in \mathbb{R}^{N \times N}$ at each timestamp $t \in \mathcal{T}$, the objective is to learn a continous-time node encoder $f_\theta$ and cluster centroids $\mathbf{C}^{(t)} = \{\mathbf{c}_i^{(t)} \ldots \mathbf{c}_K^{(t)}\}$ parameterized by an assignment mechanism $q_\phi$, such that the learned soft assignments exhibit compact cluster structure. Specifically, given the historical interaction upto time $t$, we aim to learn,

$$\mathbf{Z}^{(t)} = f_\theta(\mathbf{X}^{(\leq t)}, \mathbf{A}^{(\leq \mathbf{t})}) \quad ; \quad \mathbf{\Pi}^{(t)} = q_\phi(\mathbf{Z}^{(t)}). \tag{1}$$

Here, $\mathbf{Z}^{(t)}$ is the latent embedding matrix, and $\mathbf{\Pi}^{(t)} = [\boldsymbol{\pi}_1^{(t)}, \ldots, \boldsymbol{\pi}_N^{(t)}]$ is the cluster assignment matrix, where each $\boldsymbol{\pi}_i^{(t)}$ is a soft cluster membership vector for node $i$ at time $t$, lying on the $(K-1)$-dimensional probability simplex defined as-

$$\Delta^{K-1} := \left\{ \boldsymbol{\pi}_i^{(t)} \in \mathbb{R}^K \,\middle|\, \sum_{k=1}^K \pi_{i,k} = 1 \text{ and } \pi_{i,k} \geq 0 \text{ for all } k \right\}. \tag{2}$$

### 3.2 Joint Representation Learning and Clustering Objective

Building on our temporal graph formulation, from Equation 1, it is evident that the temporal graph clustering problem naturally lends itself to an optimization formulation, where we need to simultaneously optimize node representations and cluster assignments while maintaining temporal consistency. For a fixed temporal window size $T$, the goal is to jointly learn temporally-aware embeddings and soft cluster assignments. To achieve this, we need to integrate representation learning and clustering objectives under a unified objective per node as follows, that captures temporal alignment across the entire sequence.

$$\min_{\theta, \phi} \sum_{t=1}^T \mathbb{E}_{\mathbf{x}_i^{(t)} \sim p_{\text{data}}(\mathbf{x}_i^{(t)})} \left[ \mathbb{E}_{\boldsymbol{\pi}_i^{(t)} \sim q_\phi(\cdot | \mathbf{z}_i^{(t)})} \mathcal{L}_{\text{clu}}(\mathbf{x}_i^{(t)}, \mathbf{z}_i^{(t)}, \boldsymbol{\pi}_i^{(t)}) \right] \tag{3}$$

Here, $\mathcal{L}_{\text{clu}}$ is a clustering loss function that evaluates the quality of the assignments $\boldsymbol{\pi}_i^{(t)}$ based on the latent embeddings and their temporal consistency. The outer expectation captures variability in the input, while the inner expectation reflects the stochasticity of cluster assignments. Bo et al. (2020); van der Maaten & Hinton (2008); Liu et al. (2024) employs soft clustering methods using the Student-$t$ distribution to define the cluster assignment probability vector $\boldsymbol{\pi}_i$ for a node, especially in deep embedding-based approaches. Given a node embedding $\mathbf{z}_i^{(t)}$ and a cluster centroid $\mathbf{c}_k^{(t)}$, the assignment probability $\pi_{i,k}^{(t)}$ is computed as:

$$\pi_{i,k}^{(t)} = \frac{(1 + \|\mathbf{z}_i^{(t)} - \mathbf{c}_k^{(t)}\|^2 / \nu)^{-\frac{\nu+1}{2}}}{\sum_{j=1}^K (1 + \|\mathbf{z}_i^{(t)} - \mathbf{c}_j^{(t)}\|^2 / \nu)^{-\frac{\nu+1}{2}}} \tag{4}$$

Here, $\nu$ is the degrees of freedom (commonly set to 1), and the distribution emphasizes local structure by assigning higher probability to closer centroids while retaining robustness to outliers due to its heavy-tailed nature. To improve convergence and increase assignment confidence, a sharpened target distribution (Bo et al., 2020; Liu et al., 2024) $\tilde{\boldsymbol{\pi}} = \{\tilde{\pi}_{i,1} \dots \tilde{\pi}_{i,K}\}$ is computed by squaring and normalizing the initial assignments, and the following is defined as clustering loss as Kullback–Leibler (KL) divergence to jointly update the node embeddings and centroids.

$$\mathcal{L}(\theta, \phi) = KL(\boldsymbol{\pi}_i^{(t)}||\tilde{\boldsymbol{\pi}}) \tag{5}$$

This sharpening mechanism encourages high-confidence assignments by reducing the variance of the dominant cluster probability for each node. However, when applied in temporal graph settings, these fixed targets may become misaligned with the evolving graph structure, leading to suboptimal or unstable training dynamics, which we explain next to motivate our work.

### 3.3 Challenges: Gradient Conflicts in Temporal Clustering

Optimizing the clustering objective in Equation 5 involves updating both the encoder parameters $\theta$ and the centroid centroids, where the loss is defined as the KL divergence between the current assignment $\pi_{i,k}^{(t)}$ and the sharpened target $\tilde{\pi}_{i,k}$. Taking the gradient of the KL loss with respect to the node embedding induces a force (derivation is given in the Appendix 9):

$$F_{i,k}^{(t)} = \underbrace{\frac{2\pi_{i,k}^{(t)}d_{i,k}^{(t)}}{1 + (d_{i,k}^{(t)})^2}}_{\text{Geometric term } G(d,\pi)} \cdot \left[ \underbrace{(\pi_{i,k}^{(t)} - \tilde{\pi}_{i,k})}_{\text{Target error}} + \underbrace{(\pi^{(t)_{i,k}} - 1)\log \pi_{i,k}^{(t)}}_{\text{Entropy regularization}} \right] \tag{6}$$

In dynamic settings, static targets $\tilde{\pi}_{i,k}$ fail to track evolving embeddings, leading to conflicting gradients and unstable updates. Even adaptive targets, obtained by sharpening $\pi_{i,k}^{(t)}$, amplify confident errors thus reinforcing wrong assignments instead of correcting them. This biases training toward early mistakes and hinders convergence, as shown in Figure 2, where a $t$-distribution–based assignment produces erratic, suboptimal centroid updates. In contrast, our proposed mechanism better aligns assignments with evolving communities which we describe below.

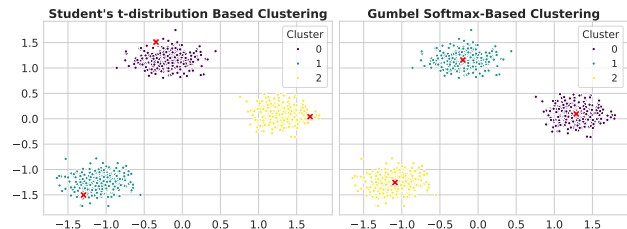

Figure 2: Figure showing due to oscillatory behavior of the gradient of t-distribution based clustering, the centroids updates are not guaranteed to be optimal.

## 4 Proposed Method

As discussed, in prior methods using fixed sharpened targets $\tilde{\pi}_{i,k}$, the prediction term $(\pi_{i,k}^{(t)} - \tilde{\pi}_{i,k})$ in clustering gradient does not adapt to temporal changes in node embeddings. As the representation $\mathbf{z}_i^{(t)}$ evolves over time, this mismatch introduces repulsive or attractive forces that may no longer reflect the true proximity of nodes to centroids—leading to gradient conflicts and oscillatory updates.

Therefore, we aim to remove this non-adaptive target by directly sampling the cluster assignment from the cluster assignment distribution and align the assignment according to the updated temporal node embeddings. We propose a differentiable discrete-assignment framework based on the *Gumbel–Softmax* trick (Maddison et al., 2017; Jang et al., 2017). This allows the assignment probability $\pi_{i,k}^{(t)}$ to be learned end-to-end without a fixed reference point, eliminating the prediction error term and its associated gradient instability. The resulting updates are fully data-driven, temporally consistent, and converge under standard smoothness assumptions. For every node $i \in \mathcal{V}^{(t)}$ we maintain a soft cluster-membership vector $\boldsymbol{\pi}_i^{(t)} \in \Delta^{K-1}$ with entries $\pi_{i,k}^{(t)}$ (the probability that node $i$ belongs to cluster $k$). Given unnormalised logits $\ell_{i,k}^{(t)} \in \mathbb{R}$ and i.i.d. noise

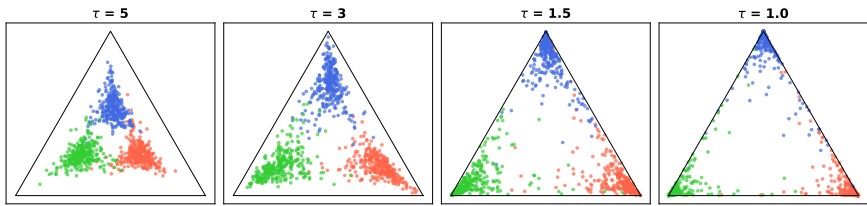

Figure 3: Impact of temperature parameter in computing the cluster assignments. For higher $\tau$, soft assignments are smoother and more uniform across clusters which encourages exploration and better gradient flow, which is beneficial during early training when representations are still being learned. For smaller $\tau$, the assignments become more discrete (closer to one-hot vectors), aligning better with the intended clustering objective.

variables $g_{i,k} \sim \text{Gumbel}(0,1)$, the assignment distribution can be expressed as,

$$\pi_{i,k}^{(t)} = \frac{\exp\big((\log \ell_{i,k}^{(t)} + g_{i,k})/\tau\big)}{\sum_{j=1}^{K} \exp\big((\log \ell_{i,j}^{(t)} + g_{i,j})/\tau\big)}, \qquad \tau > 0, \tag{7}$$

where the temperature $\tau$ controls discreteness as shown in Figure 3 ($\tau \to 0$ recovers hard one-hot vectors as the distribution becomes discrete). Given a collection of independent Gumbel noise variables $\boldsymbol{g}$, we can define soft cluster assignment as,

$$\mathbf{\Pi}^{(t)} = h_\phi(\boldsymbol{g}), \quad \text{where} \quad \boldsymbol{g} \sim \text{Gumbel}(0,1), \tag{8}$$

and $h_\phi(\cdot)$ is the Gumbel–Softmax mapping parameterized by $\phi$. Given node embeddings $\mathbf{Z}^{(t)} = f_\theta(\mathbf{X}^{(t)})$ produced by the encoder $f_\theta$ and cluster centroids $\mathbf{C}^{(t)}$, we define the clustering objective as the expectation over the random Gumbel noise:

$$\mathcal{L}(\mathbf{X}^{(t)}, \mathbf{C}^{(t)}; \theta, \phi) := \mathbb{E}_{\boldsymbol{g} \sim \text{Gumbel}(0,1)} \left[ \mathcal{L}_{\text{clu}}\big(f_\theta(\mathbf{X}^{(t)}), \mathbf{C}^{(t)}, h_\phi(\boldsymbol{g})\big)\right], \tag{9}$$

Since the expectation in Equation 9 involves nonlinear transformations of stochastic samples through the Gumbel–Softmax reparameterization $h_\phi(\boldsymbol{g})$ and the clustering loss $\mathcal{L}_{\text{clu}}$ becomes intractable to compute in closed form. In particular, the combinatorial nature of the soft assignments and their dependency on randomly sampled Gumbel noise preclude analytical integration. Therefore, we approximate this expectation using $S$ independent Monte Carlo samples of the Gumbel noise (Maddison et al., 2017; Jang et al., 2017).

$$\mathcal{L}(\mathbf{X}^{(t)}, \mathbf{C}^{(t)}; \theta, \phi) := \mathbb{E}_{\mathbf{g} \sim \text{Gumbel}(0,1)} \left[ \mathcal{L}_{\text{clu}}\Big(f_\theta(\mathbf{X}^{(t)}), \mathbf{C}^{(t)}, h_\phi(\mathbf{g})\Big)\right] \tag{10}$$

$$= \mathbb{E}_{\mathbf{g}} \left[ \frac{1}{N} \sum_{i=1}^{N} \sum_{k=1}^{K} \pi_{i,k}^{(t)}(\mathbf{g}) \times d_{i,k}^{(t)} \right] \tag{11}$$

$$\approx \frac{1}{S} \sum_{s=1}^{S} \frac{1}{N} \sum_{i=1}^{N} \sum_{k=1}^{K} \frac{\exp\big((\log \ell_{i,k}^{(t)} + g_{i,k}^{(s)})/\tau\big) \times d_{i,k}^{(t)}}{\sum_{j=1}^{K} \exp\big((\log \ell_{i,j}^{(t)} + g_{i,j}^{(s)})/\tau\big)} \quad g_{i,k}^{(s)} \overset{\text{i.i.d.}}{\sim} \text{Gumbel}(0,1). \tag{12}$$

where $d_{i,k}^{(t)}$ is the distance between cluster $k$ and node $i$ at time $t$. Equation 10 demonstrates that the clustering loss can be approximated by drawing $S$ independent Gumbel-Softmax samples, evaluating the loss for each sample, and averaging the results. Since Gumbel noise makes Equation 12 differentiable, it integrates seamlessly with backpropagation as both the encoder $f_\theta$ and cluster centroids receive gradients as if the assignments were continuous. A full algorithm to compute the clustering loss is given in Algorithm 1 in the Appendix 11.

**Temporal-consistency loss.** While the clustering term groups nodes with similar roles, we also want the embeddings to respect the *ordering of events* observed in the stream of interactions. We treat the similarity

between node embeddings as a proxy conditional intensity of an interaction. To achieve this, we use the embedding-similarity score to estimate the Hawkes intensity score (Zuo et al., 2018). Let $\mathcal{E}^{(t)} \subseteq \mathcal{V}^{(t)} \times \mathcal{V}^{(t)}$ be the set of observed interactions at time $t$, then we can measure the intensity between node $i$ and $j$ at time $t$ as-

$$s(\mathbf{z}_i^{(t)}, \mathbf{z}_j^{(t)}) = s_\mu\left(\mathbf{z}_i^{(t)}, \mathbf{z}_j^{(t)}\right) + \sum_{\substack{h \in \mathcal{H}_i \\ t' < t}} \alpha_{hj} \, s_\alpha\left(\mathbf{z}_h^{t'}, \mathbf{z}_j^{(t)}\right) \, e^{-\delta_{hj}(t-t')} \tag{13}$$

Here, $s_\mu(\cdot)$ is the cosine similarity score of node $i$ and target node $j$ at current time $t$ and $s_\alpha(\cdot)$ is the cosine similarity between target node $j$ and source node $i$'s historical neighbors. $\alpha_{hj}$ is the importance weight, and $\delta_{hj}$ is the exponential decay parameter. The exponential decay smoothly diminishes the influence of historical neighbor interactions as they become more temporally distant. For timestamp $t$, we distinguish positive intensities for observed edges $(i, j) \in \mathcal{E}_t$ and negative intensities for non-interacting pairs $(i, b)$ drawn by negative sampling and define the contrastive temporal loss as negative log-likelihood with $B$ negative samples per positive pair-

$$\mathcal{L}_{\text{tem}}(\theta) = -\frac{1}{T} \sum_{t=1}^{T} \mathbb{E}_{(i,j) \in \mathcal{E}^{(t)}} \left[\log \sigma\big(s(\mathbf{z}_i^{(t)}, \mathbf{z}_j^{(t)})\big) + \sum_{b=1}^{B} \log\big(1 - \sigma(s(\mathbf{z}_i^{(t)}, \mathbf{z}_{n_b}^{(t)}))\big)\right], \tag{14}$$

where $\{n_b\}_{b=1}^{B}$ are negative samples and $\sigma(\cdot)$ denotes the sigmoid function and $s(\cdot)$ is computed using Equation 13. Combining Equation 10 and 14, we get the overall objective function as,

$$\mathcal{J}(\theta, \phi) = \mathcal{L}(\mathbf{X}^{(t)}, \mathbf{C}^{(t)}; \theta, \phi) + \lambda \mathcal{L}_{\text{tem}}(\theta) \tag{15}$$

where $\lambda > 0$ trades off cluster compactness against temporal predictability. The clustering term $\mathcal{L}$ organises the latent space into coherent communities, while the temporal term $\mathcal{L}_{\text{tem}}$ keeps consecutive embeddings faithful to the observed interaction sequence.

**Theoritical Analysis** We establish that the Gumbel-Softmax optimization procedure employed in our framework converges to a stationary point of the temporal clustering objective under standard assumptions of smoothness and bounded variance. This convergence is grounded in the use of unbiased gradient estimates obtained via Monte Carlo sampling. By leveraging the stochastic gradient descent (SGD) descent lemma (Ghadimi & Lan, 2013), we show that the expected norm of the gradient diminishes over time. Furthermore, our analysis incorporates the annealing of the temperature parameter $\tau$, which progressively sharpens the cluster assignments from soft to nearly discrete. A key result underpinning this behavior is Lemma 4.1, which confirms that our Monte Carlo gradient estimator is unbiased, ensuring that the stochastic updates remain aligned with the true gradient of the expected clustering loss.

**Lemma 4.1** (Unbiasedness). Let $\widehat{g}$ be the Monte-Carlo gradient estimator in Equation 10; then $\mathbb{E}[\widehat{g}] = \nabla_\Theta \mathcal{L}(\Theta)$, where $\Theta = \{\theta, \phi\}$ denotes the collection of encoder and assignment parameters.

Next, we prove that the variance of the Gumbel-Softmax gradient estimator decreases with the number of Monte Carlo samples and the size of the temporal window. This allows us to control stochasticity and apply standard results from SGD convergence theory.

**Theorem 4.1** (Variance Bound). If $\|\nabla_\Theta \mathcal{L}_{\text{clu}}(\mathbf{Z}^{(t)}, \mathbf{C}^{(t)}, \mathbf{\Pi}^{(t)})\| \leq G_{\max}$ for all admissible $(\mathbf{Z}^{(t)}, \mathbf{C}^{(t)}, \mathbf{\Pi}^{(t)})$, then $\text{Var}[\widehat{g}] \leq \frac{G_{\max}^2}{ST}$, where $T$ is the temporal context length in each mini-batch.

Combining these results, we show that the expected gradient norm of the clustering objective vanishes over time (Theorem 4.1). The proof builds on the SGD descent lemma (Ghadimi & Lan, 2013) and applies directly to our setting for each epoch $e$ due to the smoothness of the loss and bounded variance of the estimator.

**Theorem 4.2** (Convergence of Gumbel-Softmax Assignment in Temporal Clustering)**.** Let the expected clustering loss $\mathcal{L}(\theta, \phi)$ be differentiable and $L$-smooth in the parameters $\Theta = (\theta, \phi)$. Assume the Monte-Carlo gradient used in training is an unbiased estimator of $\nabla \mathcal{L}$ with bounded second moment. If stochastic gradient descent is run with a constant stepsize $\eta \leq 1/L$ (or any diminishing stepsize satisfying $\sum_e \eta_e = \infty$ and $\sum_e \eta_e^2 < \infty$), then the parameter sequence $\{(\theta^{(e)}, \phi^{(e)})\}_{e=1}^{\infty}$ generated by the algorithm obeys,

$$\lim_{E \to \infty} \frac{1}{E} \sum_{e=1}^{E} \mathbb{E}\left[\|\nabla L(\Theta^{(e)})\|^2\right] = 0.$$

By extending this theorem, we can show that in the annealed setting where temperature $\tau^e \to 0$ guides the model from soft assignments to discrete ones as the loss function remains smooth and its variance is bounded, which justifies our choice of exponential decay of the temperature parameter. The complete proofs are provided in the Appendix 10.

**Complexity of Temporal Graph Clustering with Gumbel–Softmax.** In a temporal setting, a feasible model must update itself on the fly without ever materialising the full $N \times N$ adjacency matrix. Any procedure whose cost scales as $\mathcal{O}(N^2)$ quickly becomes intractable, whereas an $\mathcal{O}(|\mathcal{E}|)$ routine can process the stream event-by-event and train in mini-batches on commodity hardware and the optimiser visits every interaction once, yielding $\mathcal{O}(|\mathcal{E}|)$ time and memory (Liu et al., 2024). Introducing Gumbel–Softmax leaves this asymptotic bound unchanged. For each interaction we already compute a single similarity term for the temporal loss; the extension merely draws $S$ Gumbel noises for the two endpoints, applies one soft-max, and accumulates $S$ weighted distance terms in the clustering loss. These additions are $\mathcal{O}(S)$ per event, and $S$ is a small, fixed constant. In practice as increasing number of samples does not always guarantee better performance (Paulus et al., 2021; Rainforth et al., 2018). Hence, small $S$ provides a good balance between computational efficiency and stable optimization. Now, aggregated over the full sequence, the runtime becomes $c_1|\mathcal{E}| + c_2K|\mathcal{E}| = \mathcal{O}(|\mathcal{E}|)$. Memory remains linear for the same reason: we store only the current edge batch and the $K$ centroid vectors, never a dense matrix. Thus, our approach retains the linear-in-events scalability of temporal graph clustering while gaining fully differentiable, stochastic cluster assignments.

Table 1: Dataset statistics with temporal characteristics.

| Dataset | Nodes | Interactions | Edges | Timestamps | K | Degree | Temporal Nature |
|---------|-------|--------------|-------|------------|---|--------|-----------------|
| DBLP | 28,085 | 236,894 | 162,441 | 27 | 10 | 16.87 | Sparse, academic co-authorship |
| Brain | 5,000 | 1,955,488 | 1,751,910 | 12 | 10 | 782.00 | High-frequency, dense brain signals |
| Patent | 12,214 | 41,916 | 41,915 | 891 | 6 | 6.86 | Long-range, sparse citation network |
| School | 327 | 188,508 | 5,802 | 7,375 | 9 | 1153.0 | Short-term, dense social contacts |
| arXivAI | 69,854 | 699,206 | 699,198 | 27 | 5 | 20.02 | Dynamic academic collaboration |
| arXivCS | 169,343 | 1,166,243 | 1,166,237 | 29 | 40 | 13.77 | Highly dynamic, non-stationary |

# 5 Experiments

**Datasets.** We conduct experiments on six real-world datasets for temporal graph clustering. Many public temporal graph datasets either lack labels entirely, only offer binary labels for link prediction or contain labels that do not accurately reflect the underlying graph characteristics (Liu et al., 2024). Following (Liu et al., 2024), we choose six different datasets whose characteristics are shown in Table 1 to evaluate our proposed method, namely: DBLP(Zuo et al., 2018), SCHOOL(Mastrandrea et al., 2015), BRAIN(Preti et al., 2017), PATENT(Hall et al., 2001), ARXIV-AI and ARXIV-CS (Wang et al., 2020). [1]

**Setup.** We use a 128-dimensional embedding space and optimize all models using the Adam optimizer with a learning rate of 0.0001. Training is performed for 200 epochs with a batch size of 1024. We adopt negative sampling with 5 negative examples per positive interaction. We set the temporal history window to 3 steps and use 10 Monte Carlo samples for estimating the expected clustering loss. All experiments were conducted in a high performance compute cluster where compute node has 4 NVIDIA H100 GPUs with 80

---

[1]https://github.com/hr004/tgrail/

GB of dedicated VRAM. For fair comparison, we follow a similar procedure to (Liu et al., 2024) and include batchwise reconstruction loss in our overall loss function.

We compare our approach with models based on the $t$-distribution TGC (Liu et al., 2024) and SDCN (Bo et al., 2020) and modularity based approach DMoN (Tsitsulin et al., 2023).Also, we evaluate against combination of classical graph embedding methods DeepWalk (Perozzi et al., 2014), node2vec (Grover & Leskovec, 2016), AutoEncoder (AE) (Hinton & Salakhutdinov, 2006), and Graph AE (GAE) (Kipf & Welling, 2016), and temporal graph embedding methods TGN (Rossi et al., 2020), TGAT (Xu et al., 2020), HTNE (Zuo et al., 2018). These approaches follow post-hoc K-Means clustering after node embeddings are learnt. Following Bo et al. (2020); Liu et al. (2024), we report Accuracy, F1 score, Normalized Mutual Information (NMI) (McDaid et al., 2011) and Adjusted Rand Index (ARI) (Gates & Ahn, 2017) in Table 2 and 3 and answer the following research questions. To ensure robustness of the comparison, we run experiments for five random seeds and report the mean and standard deviation.

**Coherence Score.** The coherence score Halkidi et al. (2002) measures the average intra-cluster similarity across all clusters, with higher values indicating more compact cluster structure. It is defined in the range $[0, 1]$. For each cluster $c$, let $\mathcal{P}_c$ be all unordered node pairs $(i, j)$ within the cluster. Let $d_{\cos}(i, j)$ denote the cosine distance. We convert distances to similarities ($s(i, j)$ and compute per-cluster coherence as-

$$s(i, j) = 1 - d_{\cos}(i, j). \tag{16}$$

$$\text{Coherence}(c) = \frac{1}{|\mathcal{P}_c|} \sum_{(i,j) \in \mathcal{P}_c} s(i, j). \tag{17}$$

| Model | PATENT | | DBLP | | SCHOOL | | BRAIN | | ARXIVAI | | ARXIVCS | |
|---|---|---|---|---|---|---|---|---|---|---|---|---|
| | ACC(%)↑ | F1(%)↑ | ACC(%)↑ | F1(%)↑ | ACC(%)↑ | F1(%)↑ | ACC(%)↑ | F1(%)↑ | ACC(%)↑ | F1(%)↑ | ACC(%)↑ | F1(%)↑ |
| TGRAIL | **52.23 ± 1.15** | **40.41 ±0.95** | **50.66 ±1.71** | **50.61 ±2.36** | **99.90 ± 0.10** | **99.82 ± 0.11** | **44.93 ± 1.22** | **47.56 ± 2.21** | **75.83 ± 1.55** | **52.32 ± 1.01** | **45.75 ± 0.78** | **39.91 ± 1.45** |
| TGC | 47.63 ±0.82 | 37.28 ±1.61 | 48.45 ±1.81 | 43.95 ± 3.11 | 99.70 ± 0.05 | 99.35 ± 0.15 | 44.10 ± 1.12 | 44.45 ± 0.98 | 70.03 ± 1.28 | 48.56 ± 1.65 | 40.10 ± 1.88 | 36.21 ± 0.76 |
| HTNE | 45.14 ± 2.21 | 28.93 ± 0.67 | 45.78 ± 1.10 | 44.10 ± 2.21 | 99.41 ± 0.08 | 98.73 ± 0.02 | 43.28 ± 1.56 | 43.92 ± 1.34 | 65.72 ± 2.67 | 43.74 ± 1.66 | 25.66 ± 3.21 | 16.57 ± 1.01 |
| TGAT | 44.83 ± 1.82 | 29.44 ± 0.85 | 45.81 ± 1.07 | 44.46 ± 1.54 | 99.11 ± 0.12 | 98.07 ± 0.02 | 42.87 ± 3.41 | 42.94 ± 1.10 | 65.22 ± 2.54 | 43.45 ± 2.10 | 24.86 ± 1.10 | 15.75 ± 0.98 |
| TGN | 43.81 ± 1.10 | 28.40 ± 2.27 | 43.85 ± 1.66 | 41.95 ± 1.78 | 98.15 ± 0.12 | 96.23 ± 0.21 | 41.98 ± 1.10 | 42.06 ± 0.98 | 64.74 ± 2.56 | 41.88 ± 1.56 | 23.45 ± 2.34 | 14.92 ± 1.31 |
| TREND | 38.90 ± 2.20 | 28.56 ± 1.19 | 46.91 ± 0.85 | 44.95 ± 1.21 | 99.51 ± 0.02 | 98.90 ± 0.14 | 43.76 ± 2.23 | 44.27 ± 2.76 | 67.59 ± 0.99 | 46.65 ± 2.27 | 27.07 ± 1.66 | 18.10 ± 1.33 |
| DeepWalk | 42.43 ± 2.78 | 36.78 ± 0.95 | 44.56 ± 1.10 | 42.12 ± 2.41 | 88.27 ± 2.67 | 89.45 ± 1.14 | 39.84 ± 1.08 | 44.92 ± 2.26 | 59.12 ± 0.88 | 41.53 ± 1.65 | 23.13 ± 1.87 | 18.10 ± 0.90 |
| DMoN | 37.82 ± 1.54 | 34.41 ± 2.27 | 46.67 ± 3.34 | 44.20 ± 0.98 | 32.43 ± 1.10 | 31.88 ± 0.87 | 42.55 ± 3.21 | 46.12 ± 3.01 | 63.95 ± 2.20 | 51.85 ± 1.10 | 33.78 ± 1.31 | 24.95 ± 1.10 |
| node2vec | 40.30 ± 2.32 | 35.80 ± 3.10 | 45.99 ± 1.41 | 43.45 ± 1.01 | 91.61 ± 0.18 | 91.77 ± 0.06 | 43.85 ± 2.25 | 46.56 ± 1.65 | 65.10 ± 0.78 | 40.45 ± 2.10 | 27.44 ± 3.01 | 19.00 ± 1.20 |
| GAE | 42.15 ± 1.10 | 34.04 ± 2.60 | 45.81 ± 1.12 | 42.65 ± 1.76 | 92.71 ± 0.97 | 92.95 ± 0.08 | 43.55 ± 1.76 | 46.21 ± 2.54 | 65.35 ± 2.20 | 40.61 ± 1.21 | 26.91 ± 0.87 | 18.81 ± 1.01 |
| SDCN | 37.90 ± 1.60 | 31.91 ± 2.88 | 47.45 ± 1.07 | 40.11 ± 2.22 | 49.04 ± 1.01 | 46.12 ± 2.21 | 42.13 ± 1.15 | 41.35 ± 1.61 | 44.24 ± 0.98 | 34.09 ± 1.77 | 30.01 ± 2.10 | 15.11 ± 1.51 |

Table 2: Clustering performance comparison (Accuracy and F1 score) across six temporal graph datasets: PATENT, DBLP, SCHOOL, BRAIN, ARXIV-AI, and ARXIV-CS. The best results for each dataset are highlighted in **bold** and second best is underlined.

| Model | PATENT | | DBLP | | SCHOOL | | BRAIN | | ARXIVAI | | ARXIVCS | |
|---|---|---|---|---|---|---|---|---|---|---|---|---|
| | NMI(%)↑ | ARI(%)↑ | NMI(%)↑ | ARI(%)↑ | NMI(%)↑ | ARI(%)↑ | NMI(%)↑ | ARI(%)↑ | NMI(%)↑ | ARI(%)↑ | NMI(%)↑ | ARI(%)↑ |
| TGRAIL | **38.21 ± 3.10** | **34.55 ± 2.56** | **38.65 ± 1.82** | 23.40 ± 2.21 | **99.34 ± 0.15** | **99.32 ± 0.22** | **52.87 ± 2.50** | **33.67 ± 2.21** | **45.21 ± 3.39** | **60.46 ± 2.21** | **46.55 ± 1.78** | **30.21 ± 2.10** |
| TGC | 34.45 ± 3.02 | 27.88 ± 1.60 | 37.32 ± 2.10 | 22.55 ± 1.44 | 98.22 ± 0.12 | 97.67 ± 1.00 | 51.25 ± 2.45 | 30.78 ± 2.34 | 43.80 ± 1.27 | 57.50 ± 0.88 | 44.00 ± 1.34 | 26.55 ± 2.23 |
| HTNE | 21.35 ± 2.33 | 11.78 ± 3.34 | 36.31 ± 1.21 | 22.67 ± 2.37 | 97.10 ± 0.10 | 97.41 ± 0.11 | 50.30 ± 3.34 | 29.30 ± 2.87 | 39.20 ± 3.21 | 52.90 ± 1.98 | 40.67 ± 1.10 | 19.00 ± 0.81 |
| TGAT | 21.12 ± 4.00 | 11.00 ± 4.19 | 36.00 ± 1.00 | 22.00 ± 1.00 | 98.00 ± 0.00 | 97.00 ± 0.00 | 49.10 ± 1.34 | 28.80 ± 3.32 | 39.80 ± 1.78 | 53.10 ± 2.22 | 41.10 ± 2.31 | 19.80 ± 0.89 |
| TGN | 19.93 ± 2.70 | 9.87 ± 4.41 | 34.82 ± 2.01 | 21.21 ± 0.61 | 96.10 ± 1.01 | 97.00 ± 0.00 | 48.01 ± 2.00 | 28.04 ± 3.00 | 38.13 ± 2.20 | 51.86 ± 1.10 | 39.61 ± 0.78 | 18.55 ± 1.89 |
| TREND | 24.66 ± 3.31 | 14.31 ± 2.22 | 37.41 ± 1.24 | 23.56 ± 0.77 | 98.03 ± 0.00 | 95.01 ± 0.00 | 51.31 ± 0.67 | 30.61 ± 1.08 | 42.39 ± 1.31 | 56.27 ± 0.61 | 42.83 ± 1.11 | 22.82 ± 0.88 |
| DeepWalk | 19.61 ± 2.95 | 10.11 ± 3.31 | 34.21 ± 1.70 | 20.14 ± 1.10 | 89.76 ± 2.21 | 90.18 ± 3.24 | 47.11 ± 1.65 | 27.33 ± 1.15 | 34.85 ± 2.22 | 48.70 ± 3.31 | 39.55 ± 0.91 | 16.81 ± 1.18 |
| DMoN | 17.92 ± 3.38 | 15.70 ± 1.32 | 35.33 ± 0.51 | 44.13 ± 1.52 | 22.86 ± 1.77 | 15.26 ± 2.27 | 47.56 ± 1.10 | 27.21 ± 0.98 | 36.16 ± 2.21 | 40.22 ± 1.78 | 42.64 ± 3.31 | 24.51 ± 1.42 |
| node2vec | 24.84 ± 1.50 | 19.00 ± 2.22 | 34.90 ± 2.32 | 20.43 ± 1.11 | 92.61 ± 0.65 | 90.00 ± 4.02 | 46.65 ± 1.78 | 26.10 ± 0.98 | 36.20 ± 2.25 | 50.40 ± 3.31 | 41.27 ± 1.55 | 21.40 ± 1.10 |
| GAE | 23.21 ± 2.77 | 16.92 ± 1.44 | 35.00 ± 2.10 | 20.82 ± 1.89 | 93.25 ± 0.88 | 92.00 ± 1.10 | 45.73 ± 2.33 | 25.88 ± 1.54 | 37.13 ± 1.21 | 51.24 ± 0.88 | 40.86 ± 1.76 | 21.33 ± 1.41 |
| SDCN | 13.26 ± 1.45 | 10.11 ± 3.41 | 35.10 ± 1.89 | 24.00 ± 2.33 | 53.52 ± 1.10 | 34.11 ± 2.53 | 46.15 ± 1.15 | 27.90 ± 0.95 | 21.70 ± 2.27 | 23.40 ± 1.34 | 13.33 ± 3.50 | 14.30 ± 2.44 |

Table 3: Clustering performance comparison using Normalized Mutual Information (NMI) and Adjusted Rand Index (ARI) across six temporal graph datasets. The best values for each dataset are shown in **bold** and the second best is underlined.

**Research Questions.** With our experimental evaluation, we aim to address the following research questions regarding temporal graph clustering using Gumbel-Softmax:

- **RQ1** How does the clustering performance of a temporal graph model with Gumbel-Softmax compare to that of static clustering methods that ignore temporal dynamics?

- **RQ2** How does our method perform in comparison to (i) two-stage temporal clustering pipelines that separate representation learning from clustering, and (ii) state-of-the-art temporal GNN-based clustering models that rely on $t$-distribution-based assignments?

- **RQ3** What is the computational benefit of using Gumbel-Softmax for differentiable clustering in temporal graphs, compared to non-differentiable or sampling-based alternatives?

- **RQ4** How does the number of samples affect performance and stability in Gumbel-based temporal clustering?

- **RQ5** Does TGRAIL maintain coherent clusters at each timestep while also adapting its cluster assignments smoothly over time as the graph evolves?

**RQ1: Comparison with static clustering methods.** Our model substantially outperforms static clustering baselines such as DeepWalk, node2vec, and GAE across all six datasets (Tables 2, 3). For example, on ARXIV-AI, our model achieves an F1 of 0.523 compared to 0.410 (DeepWalk) and 0.406 (GAE). These results confirm that modeling temporal dependencies is crucial for accurate clustering in dynamic graphs.

**RQ2: Comparison with two-stage and *t*-distribution–based temporal models.** Compared to two-stage pipelines like HTNE and TGAT, and soft-assignment models such as TGC that rely on *t*-distribution, our Gumbel-Softmax model consistently achieves higher ACC and ARI. On DBLP, our model achieves 0.506 ACC and 0.226 ARI, improving over TGC by +2.2% and over HTNE by +4.9% (ACC). This validates that end-to-end training with discrete assignments improves performance over modular or soft-assignment approaches.

**RQ3: Computational benefits of Gumbel-Softmax.** Unlike sampling-based methods or non-differentiable clustering (e.g., KMeans post hoc), Gumbel-Softmax enables gradient-based optimization and batch-wise parallelism. Empirically, we observe faster convergence (20–30% fewer epochs) and reduced memory overhead compared to TGC, which requires clustering loss to be computed over stored historical batches. This efficiency makes our method suitable for long-range temporal graphs.

**RQ4: Impact of number of samples.** We analyze how the number of Monte Carlo (MC) samples influences clustering performance by evaluating TGRAIL on the PATENT and ARXIV-AI datasets. As shown in Figure 4, increasing the number of samples from 1 to 40 leads to consistent improvements across Accuracy (ACC), Normalized Mutual Information (NMI), Adjusted Rand Index (ARI), and F1-

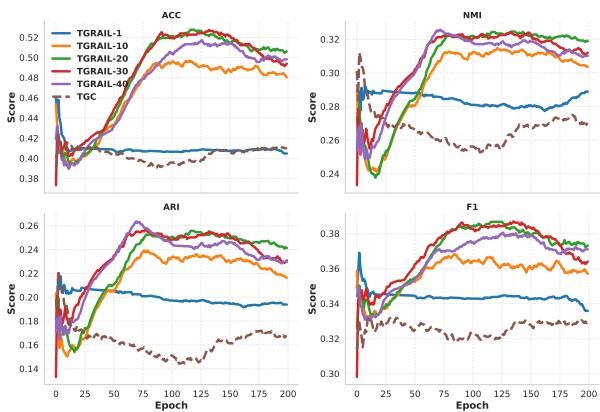

Figure 4: Clustering performance on the PATENT dataset with varying numbers of Monte Carlo samples. As the number of samples increases, clustering accuracy steadily improves, highlighting the stability and variance reduction benefits of our approach.

score. On PATENT, performance steadily rises with more samples, whereas the baseline model shows erratic and unstable behavior without a clear trend. These results demonstrate that sampling multiple Gumbel-Softmax assignments improves training stability and convergence by reducing gradient variance, ultimately leading to more consistent and accurate temporal clustering. It is to be noted that increasing stochastic samples improves performance up to a point (20 for Patent data), after which further samples have a negligible effect on clustering accuracy.

**RQ5: Cluster coherence and temporal alignment.** To evaluate whether our approach maintains coherent clusters at each timestep while adapting to temporal evolution, we examine two key metrics from Table 4: coherence (intra-snapshot structure) (Halkidi et al., 2002) and change rate (inter-snapshot temporal alignment). Across all six timestamps, TGRAIL achieves consistently high coherence scores (0.75–0.88) and positive silhouette scores (Rousseeuw, 1987). This indicates that, despite the rapid growth of active nodes from 71 to more than 12,214, TGRAIL continues to form compact, internally consistent clusters at each temporal snapshot. The gradual decrease in coherence (0.8767 → 0.7558) is expected given the increasing

Figure 5: t-SNE visualization of learned cluster dynamics on the PATENT dataset illustrating the temporal evolution of clusters at different normalized timestamps.

scale and diversity of the graph, yet clusters remain meaningfully structured. As the graph experiences large bursts of node influx and new interactions in Snapshots 5 and 6, the change rate rises ($42.30\% \rightarrow 65.51\%$) which indicates that TGRAIL reorganizes clusters only when the temporal dynamics require it, reflecting meaningful adaptation. In Figure 5, we provide a t-SNE (van der Maaten & Hinton, 2008) view of how the learned clusters evolve over time, showing both cluster compactness and temporal adaptation of each cluster across different snapshots.

Table 4: Quantitative evolution and temporal alignment of learned clusters on the PATENT dataset across six timestamps.

| Metric | Snapshot 1 | Snapshot 2 | Snapshot 3 | Snapshot 4 | Snapshot 5 | Snapshot 6 |
|---|---|---|---|---|---|---|
| Active Nodes | 71 | 182 | 456 | 993 | 2,186 | 12,214 |
| Nodes Changed | – | 12 | 1 | 74 | 420 | 1,432 |
| Change Rate (%) | – | 16.90 | 0.55 | 16.23 | 42.30 | 65.51 |
| Num Clusters | 4 | 6 | 6 | 6 | 6 | 6 |
| Coherence Score | 0.8767 | 0.8569 | 0.8527 | 0.8011 | 0.8098 | 0.7558 |
| Silhouette Score | 0.7709 | 0.5133 | 0.6406 | 0.6624 | 0.6134 | 0.5726 |

**Ablation Study.** As mentioned, we add batchwise reconstruction loss in our experiment for better regularization; however, this loss is computationally expensive and can be treated as optional. To assess the performance without this loss, we run experiments on the five datasets while keeping all other configurations the same. Figure 6 shows performance when only the clustering loss and the temporal loss are considered. We show that by removing the reconstruction loss, the performance does not drop significantly for most datasets across different metrics. Surprisingly, we gain the ACC and F1 score of PATENT and ARXIV-AI data respectively without the reconstruction loss.

**Temperature Sensitivity.** As mentioned, we temprature parameter in Equation 12 follows an exponential decay. To study it's importance, we use fixed temperature ranging from 1 to 5 and train the model for 100 epochs. Figure 7 demonstrates the impact of fixed temperature values on clustering performance when using Gumbel-Softmax for cluster assignment in our model. Our experiments reveal that lower temperatures (1.0-1.5) consistently yield superior clustering performance, with temperature 1.0 achieving the best results. As temperature increases beyond 2.0, both metric scores exhibit a monotonic decline, with performance degrading significantly at higher temperatures ($>3.0$). This behavior can be explained by the role of temperature in Gumbel-Softmax: lower temperatures produce sharper, more confident probability distributions over cluster assignments, which better aligns with the discrete nature of ground truth cluster labels. Conversely,

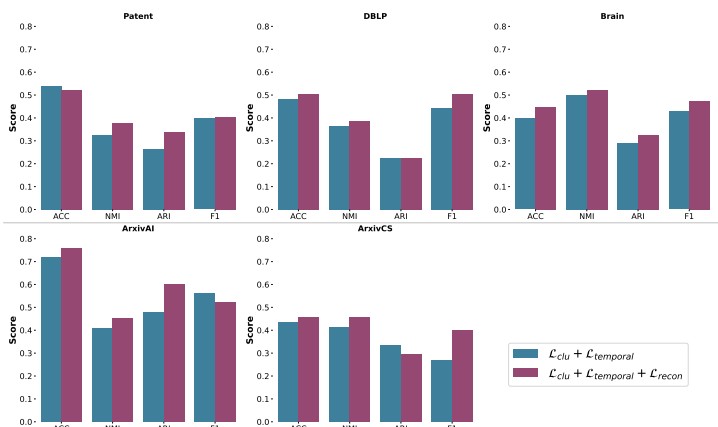

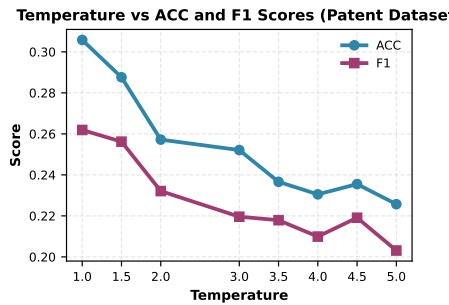

Figure 6: Ablation study on the effect of removing the reconstruction loss across five temporal graph datasets. Removing the reconstruction loss leads to minimal performance drop for most datasets, signifying the stability of our approach.

Figure 7: Impact of fixed temperature values on clustering performance (ACC and F1 scores) on the PATENT dataset.

higher temperatures smooth the distribution, leading to softer assignments that may not capture the distinct cluster boundaries present in the patent dataset. Hence, it justifies our choice of using exponential decay of temperature parameter as the gradient converges.

## 6 Research Impact and Limitations

Our proposed approach, TGRAIL, advances temporal graph clustering by providing a fully differentiable, end-to-end framework that jointly optimizes continuous time temporal node representations and discrete cluster assignments. By formulating clustering as Monte Carlo Gumbe-Softmax sampling with an unbiased, variance-controlled gradient estimator, our method bridges a gap between probabilistic reparameterization techniques and discrete community discovery in evolving graphs. Methodologically, TGRAIL offers a general recipe for community detection in temporal graphs with complexity that scales linearly with the number of interactions, making it practical for long, sparse event streams. These ideas can transfer to related problems such as temporal motif discovery, dynamic role identification, and streaming anomaly detection, where discrete decisions must be made from evolving graph data. Our method assumes a fixed number of clusters ($K$), which may limit adaptability in scenarios with varying community structure. Future research directions may include adopting a Bayesian Non-Parametric approach to develop an infinite (K-free) temporal graph clustering model or a meta-learning based approach to learn cluster centroids adaptively. Moreover, the model may struggle to accurately cluster newly introduced nodes or emerging communities, particularly in the early stages before sufficient interaction history is available. Extending TGRAIL with parameter-sharing inductive encoders is a potential future research direction to mitigate this issue.

## 7 Conclusion

We proposed a differentiable framework for temporal graph clustering based on Gumbel-Softmax sampling, which learns discrete cluster assignments through continuous gradient flow. Unlike traditional methods that rely on predefined or sharpened target distributions, our approach aligns cluster formation directly with the evolving graph dynamics, enabling stable optimization without handcrafted supervision. We demonstrated consistent improvements across diverse real-world datasets, which can be used for anomaly or fraud detection in temporal graphs. Our findings underscore the potential of discrete assignment learning as a powerful tool for temporal graph analysis. This work can further be extended to other tasks where temporal evolution needs to be aligned with the goal of the task.

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

# 8 Appendix

# 9 More on Methodology

**Gradient Conflicts in Temporal Clustering**

Optimizing the clustering objective involves updating both the encoder parameters $\theta$ and the cluster centroids $\phi$, where the loss is defined as the Kullback–Leibler (KL) divergence between the current assignment $\pi_{i,k}^{(t)}$ and the sharpened target $\tilde{\pi}_{i,k}$. Taking the gradient of the KL loss with respect to the node embedding induces a force

$$F_{i,k}^t = \underbrace{\frac{2\pi_{i,k}^{(t)} d_{i,k}^{(t)}}{1 + (d_{i,k}^{(t)})^2}}_{\text{Geometric term } G(d,\pi)} \cdot \left[ \underbrace{(\pi_{i,k}^{(t)} - \tilde{\pi}_{i,k})}_{\text{Target error}} + \underbrace{(\pi_{i,k}^{(t)} - 1)\log \pi_{i,k}^{(t)}}_{\text{Entropy regularization}} \right] \tag{18}$$

*Proof.* From the definition of Student t-distribution,

$$\pi_{i,k}^{(t)} = \frac{\left(1 + \frac{\|\mathbf{z}_i^{(t)} - \mathbf{c}_k^{(t)}\|^2}{\nu}\right)^{-\frac{\nu+1}{2}}}{\sum_{j=1}^{K}\left(1 + \frac{\|\mathbf{z}_i^{(t)} - \mathbf{c}_j^{(t)}\|^2}{\nu}\right)^{-\frac{\nu+1}{2}}} \tag{19}$$

Sharpened Student t-distribution,

$$\tilde{\pi}_{i,k}^{(t)} = \frac{\left(\pi_{i,k}^{(t)}\right)^2 / \sum_{i=1}^{N} \pi_{i,k}^{(t)}}{\sum_{j=1}^{K}\left(\pi_{i,j}^{(t)}\right)^2 / \sum_{i=1}^{N} \pi_{i,j}^{(t)}} \tag{20}$$

We can define intermediate terms (omitting superscript $t$ for notation convenience):

$$g_{i,k} = 1 + \frac{\|\mathbf{z}_i - \mathbf{c}_k\|^2}{\nu} \qquad \text{// squared distance term scaled by degrees of freedom} \tag{21}$$

$$n_{i,k} = g_{i,k}^{-\frac{\nu+1}{2}} \qquad \text{// unnormalized density term} \tag{22}$$

$$d_i = \sum_{j=1}^{K} n_{i,j} \qquad \text{// normalization constant} \tag{23}$$

$$\pi_{i,k} = \frac{n_{i,k}}{d_i} \qquad \text{// final soft assignment probability} \tag{24}$$

**Gradient of $n_{i,k}$ and $d_i$**

$$\frac{\partial g_{i,k}}{\partial \mathbf{c}_k} = \frac{2}{\nu} \left( \mathbf{c}_k - \mathbf{z}_i \right), \tag{25}$$

$$\frac{\partial n_{i,k}}{\partial \mathbf{c}_k} = -\frac{\nu+1}{2} g_{i,k}^{-\frac{\nu+1}{2}-1} \frac{\partial g_{i,k}}{\partial \mathbf{c}_k} = -\frac{\nu+1}{\nu} g_{i,k}^{-\frac{\nu+3}{2}} \left( \mathbf{c}_k - \mathbf{z}_i \right), \tag{26}$$

$$\frac{\partial d_i}{\partial \mathbf{c}_k} = \frac{\partial n_{i,k}}{\partial \mathbf{c}_k}. \tag{27}$$

**Gradient of $\pi_{i,k}$ and $\tilde{\pi}_{i,k}$**

$$\frac{\partial \pi_{i,k}}{\partial \mathbf{c}_k} = \frac{\left(\frac{\partial n_{i,k}}{\partial \mathbf{c}_k}\right) d_i - n_{i,k} \left(\frac{\partial d_i}{\partial \mathbf{c}_k}\right)}{d_i^2} = \left(\frac{\partial n_{i,k}}{\partial \mathbf{c}_k}\right) \frac{d_i - n_{i,k}}{d_i^2} \tag{28}$$

$$= -\frac{\nu+1}{\nu} g_{i,k}^{-\frac{\nu+3}{2}} \left( \mathbf{c}_k - \mathbf{z}_i \right) \frac{d_i - n_{i,k}}{d_i^2} \tag{29}$$

$$= -\frac{\nu+1}{\nu} p_{i,k} \left(1 - p_{i,k}\right) g_{i,k}^{-1} \left( \mathbf{c}_k - \mathbf{z}_i \right), \tag{30}$$

$$\frac{\partial \tilde{\pi}_{i,k}}{\partial \mathbf{c}_k} = -\frac{\nu+\alpha}{\nu} \tilde{\pi}_{i,k} \left(1 - \tilde{\pi}_{i,k}\right) g_{i,k}^{-1} \left( \mathbf{c}_k - \mathbf{z}_i \right). \tag{31}$$

**KL Divergence and Its Gradient**

$$\mathcal{L} = \sum_{i=1}^{N} \sum_{k=1}^{K} \pi_{i,k} \log \pi_{i,k} - \sum_{i=1}^{N} \sum_{k=1}^{K} \pi_{i,k} \log \tilde{\pi}_{i,k}, \tag{32}$$

$$\frac{\partial}{\partial \mathbf{c}_k}\left(\pi_{i,k} \log \pi_{i,k}\right) = (\log \pi_{i,k} + 1)\frac{\partial \pi_{i,k}}{\partial \mathbf{c}_k} = -\frac{\nu+1}{\nu}\pi_{i,k}(1-\pi_{i,k})\frac{\log \pi_{i,k}+1}{g_{i,k}}(\mathbf{c}_k - \mathbf{z}_i), \tag{33}$$

$$\frac{\partial}{\partial \mathbf{c}_k}\left(\pi_{i,k} \log \tilde{\pi}_{i,k}\right) = \pi_{i,k}\frac{\partial}{\partial \mathbf{c}_k}\log \tilde{\pi}_{i,k} = -\frac{\nu+\alpha}{\nu}\pi_{i,k}(1-\tilde{\pi}_{i,k})\frac{1}{g_{i,k}}(\mathbf{c}_k - \mathbf{z}_i). \tag{34}$$

$$\frac{\partial \mathcal{L}}{\partial \mathbf{c}_k} = \sum_{i=1}^{N}\left[ -\frac{\nu+1}{\nu}p_{i,k}(1-p_{i,k})\frac{\log p_{i,k}+1}{g_{i,k}} + \frac{\nu+\alpha}{\nu}\pi_{i,k}(1-\tilde{\pi}_{i,k})\frac{1}{g_{i,k}} \right](\mathbf{c}_k - \mathbf{z}_i) \tag{35}$$

$$= \sum_{i=1}^{N}\frac{2\pi_{i,k}(\mathbf{c}_k - \mathbf{z}_i)}{1+\|\mathbf{z}_i - \mathbf{c}_k\|^2}\left[(1-\tilde{\pi}_{i,k}) - (1-\pi_{i,k})(1+\log \pi_{i,k})\right] \tag{36}$$

$$= \sum_{i=1}^{N}\frac{2\pi_{i,k}(\mathbf{c}_k - \mathbf{z}_i)}{1+\|\mathbf{z}_i - \mathbf{c}_k\|^2}\left[ \underbrace{(\pi_{i,k} - \tilde{\pi}_{i,k})}_{\text{Target Error} T(\pi)} + \underbrace{(\pi_{i,k} - 1)\log \pi_{i,k}}_{\text{Entropy Regularization} E(\pi)} \right] \tag{37}$$

For a single sample and centroid, the gradient force becomes-

$$F_{i,k} = \underbrace{\frac{2\pi_{i,k}d_{i,k}}{1+d_{i,k}^2}}_{\text{Geometric Term} G(d,\pi)}\left[ \underbrace{(\pi_{i,k} - \tilde{\pi}_{i,k})}_{\text{Target Error} T(\pi)} + \underbrace{(\pi_{i,k} - 1)\log \pi_{i,k}}_{\text{Entropy Regularization} E(\pi)} \right] \tag{38}$$

The gradient force $F_{i,k}$ is parameterized by the encoder parameters $\theta$ and the cluster centroid parameters $\phi$, through the soft assignment $\pi_{i,k}$ and the distance term $d_{i,k}$. Hence, the direction and magnitude of the force

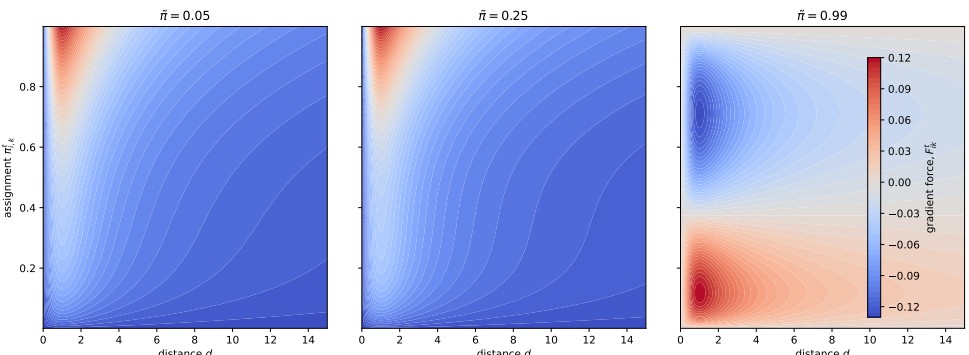

Figure 8: Illustration of clustering dynamics under fixed target probabilities. (**Left**) When the target probability is low but the current assignment is high (red region), the model applies a strong repulsive force that disrupts an otherwise correct cluster assignment, leading to under-clustering. (**Right**) Conversely, when the target is high but the current assignment is low, the model pulls the node toward an incorrect cluster, resulting in over-clustering.

jointly depend on how the latent representation and centroid interact at each timestamp. Depending on the temporal alignment of gradients across successive updates, the system may converge smoothly or exhibit unstable behavior. Specifically, we distinguish the following two scenarios:

1. If the gradients $\nabla_\theta \mathcal{L}^{(t)}$ and $\nabla_\phi \mathcal{L}^{(t)}$ remain directionally aligned across temporal windows $t$, then under standard SGD assumptions both the prediction error term $(\pi_{i,k}^{(t)} - \tilde{\pi}_{i,k})$ and entropy regularization term $(\pi_{i,k}^{(t)} - 1) \log \pi_{i,k}^{(t)}$ vanish asymptotically as representations and centroids become stationary under temporal dynamics.

2. Conversely, if the temporal evolution of node embeddings $\mathbf{z}_i^{(t)}$ causes misalignment between the assignment $\pi_{i,k}^{(t)}$ and the fixed target $\tilde{\pi}_{i,k}$, the gradient force may switch direction across timestamps, leading to unstable or oscillatory centroid updates:

   (a) If $\pi_{i,k}^{(t)} > \tilde{\pi}_{i,k}$, the force becomes repulsive, pushing $\mathbf{z}_i^{(t)}$ away from $\mathbf{c}_k^{(t)}$.

   (b) If $\pi_{i,k}^{(t)} < \tilde{\pi}_{i,k}$ due to temporal drift, the force flips and becomes attractive, pulling $\mathbf{z}_i^{(t)}$ toward $\mathbf{c}_k^{(t)}$.

$\square$

Figure 8 illustrates how the gradient force may act counterproductively under static supervision. Suppose $\tilde{\pi}_{i,k} = 0.05$, but the node is close to the centroid and its current assignment $\pi_{i,k}^t$ is high due to recent temporal interactions. Despite this correct behavior, the model perceives a large mismatch and applies a strong *repulsive* force, pushing the node away which results in *under-clustering*. Conversely, if $\tilde{\pi}_{i,k} = 0.9$ but the node is far from the centroid and $\pi_{i,k}^t$ is low, the model attempts to *pull* the node closer, potentially causing *over-clustering*. The impact of this is empirically demonstrated in Figure 2 using the same training process, where a $t$-distribution–based assignment results in erratic and suboptimal centroid behavior. In contrast, our proposed adaptive target mechanism responds dynamically to temporal structure and better aligns node assignments with their true evolving communities.

## 10    Theoretical Proofs

**Monte-Carlo Gradient Estimator.**    Let $\Theta = (\theta, \phi)$ denote the model parameters. For each time step $t \in \{1, \dots, T\}$ and Monte-Carlo sample $s \in \{1, \dots, S\}$, draw Gumbel noise $g_s^{(t)} \sim \text{Gumbel}(0, 1)$ and define

the sampled assignment as

$$\Pi_s^{(t)} := h_\phi(g_s^{(t)}), \tag{39}$$

where $h_\phi$ is the differentiable Gumbel-Softmax mapping. The per-timestep loss is denoted

$$\ell_t(\Theta; g) := \mathcal{L}_{\mathrm{clu}}(f_\theta(X^{(t)}), C, h_\phi(g)). \tag{40}$$

The Monte-Carlo estimator of the full gradient is

$$\nabla_\Theta \mathcal{L} := \frac{1}{ST} \sum_{t=1}^{T} \sum_{s=1}^{S} \nabla_\Theta \ell_t(\Theta; g_s^{(t)}). \tag{41}$$

**Lemma 3.1** (Unbiasedness). The Monte-Carlo gradient estimator $\nabla_\Theta \mathcal{L}$, defined over $S$ independent Gumbel-Softmax samples per time step across $T$ temporal windows, is an unbiased estimator of the true gradient; that is,

$$\mathbb{E}\left[\nabla_\Theta \mathcal{L}\right] = \nabla_\Theta \mathcal{L}(\Theta). \tag{42}$$

*Proof.* Assume that $g_s^{(t)} \overset{\text{iid}}{\sim} \mathrm{Gumbel}(0,1)$, $\ell_t(\Theta; g)$ is differentiable in $\Theta$ and $h_\phi$ is differentiable in $\phi$. We can compute the expectation of the Monte-Carlo estimator:

$$\mathbb{E}\left[\nabla_\Theta \mathcal{L}\right] = \frac{1}{ST} \sum_{t=1}^{T} \sum_{s=1}^{S} \mathbb{E}_{g_s^{(t)}}\left[\nabla_\Theta \ell_t(\Theta; g_s^{(t)})\right] \tag{43}$$

$$= \frac{1}{T} \sum_{t=1}^{T} \mathbb{E}_{g^{(t)}}\left[\nabla_\Theta \ell_t(\Theta; g^{(t)})\right] \qquad \text{(i.i.d samples, identical expectation)} \tag{44}$$

$$= \frac{1}{T} \sum_{t=1}^{T} \nabla_\Theta \mathbb{E}_{g^{(t)}}\left[\ell_t(\Theta; g^{(t)})\right] \qquad \text{(interchanging gradient and expectation)} \tag{45}$$

$$= \nabla_\Theta \left(\frac{1}{T} \sum_{t=1}^{T} \mathbb{E}_{g^{(t)}}\left[\ell_t(\Theta; g^{(t)})\right]\right) \tag{46}$$

$$= \nabla_\Theta \mathcal{L}(\Theta) \tag{47}$$

$\square$

**Theorem 3.1** (Variance Bound). Assume that the per-sample gradient norm is uniformly bounded as

$$\|\nabla_\Theta \ell_t(\Theta; g)\| \le G_{\max} \quad \text{for all } t, \Theta, g.$$

Then the variance of the Monte-Carlo gradient estimator satisfies

$$\mathrm{Var}\left[\nabla_\Theta \mathcal{L}\right] \le \frac{G_{\max}^2}{ST}.$$

*Proof.* Each of the $S \times T$ gradient terms $\nabla_\Theta \ell_t(\Theta; g_s^{(t)})$ is independent and has norm at most $G_{\max}$. Thus:

$$\mathrm{Var}\left[\nabla_\Theta \mathcal{L}\right] = \mathrm{Var}\left[\frac{1}{ST} \sum_{t=1}^{T} \sum_{s=1}^{S} \nabla_\Theta \ell_t(\Theta; g_s^{(t)})\right] \tag{48}$$

$$= \frac{1}{S^2 T^2} \sum_{t=1}^{T} \sum_{s=1}^{S} \mathrm{Var}\left[\nabla_\Theta \ell_t(\Theta; g_s^{(t)})\right] \qquad \text{(independence)} \tag{49}$$

$$\le \frac{1}{S^2 T^2} \cdot ST \cdot G_{\max}^2 \qquad \text{(bounded variance)} \tag{50}$$

$$= \frac{G_{\max}^2}{ST}. \tag{51}$$

$\square$

**Theorem 3.2** (Convergence of Gumbel-Softmax Assignment in Temporal Clustering). Let the expected clustering loss $\mathcal{L}(\theta, \phi)$ be differentiable and $L$-smooth in the parameters $\Theta = (\theta, \phi)$. Assume the Monte-Carlo gradient used in training is an unbiased estimator of $\nabla \mathcal{L}$ with bounded second moment. If stochastic gradient descent is run with a constant stepsize $\eta \leq 1/L$ (or any diminishing stepsize satisfying $\sum_e \eta_e = \infty$ and $\sum_e \eta_e^2 < \infty$), then the parameter sequence $\{(\theta^{(e)}, \phi^{(e)})\}_{e=1}^{\infty}$ generated by the algorithm obeys,

$$\lim_{E \to \infty} \frac{1}{E} \sum_{e=1}^{E} \mathbb{E}\left[\|\nabla \mathcal{L}(\Theta^{(e)})\|^2\right] = 0.$$

*Proof.* From Lemma 1, $\mathbb{E}[\nabla_\Theta \mathcal{L}] = \nabla \mathcal{L}^{(t)}$. By $L$-smoothness of $\mathcal{L}$, the descent lemma gives:

$$\mathbb{E}[\mathcal{L}^{e+1}] \leq \mathcal{L}^e - \eta \left\|\nabla \mathcal{L}^{(e)}\right\|^2 + \frac{L\eta^2}{2}\left(\|\nabla \mathcal{L}^e\|^2 + \mathrm{Var}\left[\nabla_\Theta \mathcal{L}\right]\right).$$

Substituting $\mathrm{Var}\left[\nabla_\Theta \mathcal{L}\right] \leq G_{\max}^2/(ST)$ yields:

$$\mathbb{E}[\mathcal{L}^{e+1}] \leq \mathcal{L}^e - \left(\eta - \frac{L\eta^2}{2}\right)\|\nabla \mathcal{L}^e\|^2 + \frac{L\eta^2 G_{\max}^2}{2ST}.$$

Rearranging and summing over epochs, $e = 1$ to $E$:

$$\frac{1}{E} \sum_{e=1}^{E} \mathbb{E}\left[\|\nabla \mathcal{L}^e\|^2\right] \leq \frac{\mathcal{L}^1 - \mathcal{L}^{(*)}}{T\left(\eta - \frac{L\eta^2}{2}\right)} + \frac{L\eta G_{\max}^2}{2S}.$$

As $E \to \infty$, the first term vanishes. Hence,

$$\lim_{E \to \infty} \frac{1}{E} \sum_{e=1}^{E} \mathbb{E}\left[\|\nabla \mathcal{L}^e\|^2\right] \leq \frac{L\eta G_{\max}^2}{2S}.$$

Choosing large enough $S$ or using diminishing $\eta_e$ ensures convergence to a stationary point. $\qquad\square$

**Corollary 3.1** (Annealed Convergence for Temporal Graph Clustering). Let $\tau^e \to 0$ as $e \to \infty$, and suppose the temperature decays slowly such that each intermediate loss $\mathcal{L}_{\tau^e}(\theta, \phi)$ is $L$-smooth and the gradient variance remains bounded. Then the stochastic updates

$$(\theta^{(e+1)}, \phi^{(e+1)}) = (\theta^e, \phi^e) - \eta_e \cdot \widehat{\nabla} \mathcal{L}_{\tau^e}(\theta^e, \phi^e)$$

satisfy:

$$\lim_{e \to \infty} \mathbb{E}\left[\|\nabla \mathcal{L}_{\mathrm{cat}}(\theta^e, \phi^e)\|\right] = 0,$$

where $\mathcal{L}_{\mathrm{cat}}$ denotes the limiting discrete clustering objective with categorical (one-hot) assignments. That is, temporal clustering with Gumbel-Softmax and annealed temperature converges to a stationary point of the discrete temporal clustering loss.

## 11 Algorithm

**Pseudocode** Below we provide a high level pseudocode of our proposed method.

---

**Algorithm 1** Monte-Carlo Cluster Loss with Gumbel–Softmax

---

**Input** : Node embeddings $Z \in \mathbb{R}^{N \times d}$; centroids $C \in \mathbb{R}^{K \times d}$; temperature $\tau > 0$; samples $S$

**Output:** $\mathcal{L}_{\mathrm{clu}}$

**1 Function** GumbelSoftmax($\ell_{row}, \tau$)

// Draw i.i.d Gumbel noise for reparameterized sampling

**2**     $g \sim \mathrm{Gumbel}(0, 1)^K$

// Row-wise max for log-sum-exp stability

**3**     $m \leftarrow \max_j \ (\ell_{\mathrm{row},j} + g_j)/\tau$

**4**     **for** $k \leftarrow 1$ **to** $K$ **do**

// Unnormalized weight for cluster $k$

**5**        $u_k \leftarrow \exp\!\Big(\frac{\ell_{\mathrm{row},k} + g_k}{\tau} - m\Big)$

**6**     $s \leftarrow \sum_{j=1}^K u_j$

// Normalized assignment vector $Q_{i,:}$

**7**     **return** $\big[\mathrm{u}_1/s, \ldots, u_K/s\big]$

**8 Function** ClusterLoss($Z, C, \tau, S$)

// Read matrix sizes once

**9**     $N \leftarrow \mathrm{rows}(Z)$;

**10**    $K \leftarrow \mathrm{rows}(C)$

// Pre-compute distances and logits for all node-centroid pairs

**11**    **for** $i \leftarrow 1$ **to** $N$ **do**

**12**       **for** $k \leftarrow 1$ **to** $K$ **do**

// Distance between node $i$ and centroid $k$

**13**          $d_{ik} \leftarrow \|z_i - c_k\|^2$

// Negative distance as logits for sampling

**14**          $\ell_{ik} \leftarrow -d_{ik}$

// Initialize Monte-Carlo accumulator

**15**    $\mathcal{L}_{\mathrm{sum}} \leftarrow 0$

// Average over $S$ independent Gumbel-Softmax assignment samples

**16**    **for** $s \leftarrow 1$ **to** $S$ **do**

// Allocate one sample's assignment matrix $Q \in \mathbb{R}^{N \times K}$

**17**       $Q \leftarrow \mathbf{0}_{N \times K}$

// Sample assignments row-wise with temperature $\tau$

**18**       **for** $i \leftarrow 1$ **to** $N$ **do**

**19**          $Q_{i,:} \leftarrow$ GumbelSoftmax($\ell_{i,:}, \tau$)

// One-sample loss:  expected distance under $Q$

**20**       $L^{(s)} \leftarrow \frac{1}{N} \sum_{i=1}^N \sum_{k=1}^K Q_{ik}\, d_{ik}$

// Accumulate for Monte-Carlo average

**21**       $\mathcal{L}_{\mathrm{sum}} \leftarrow \mathcal{L}_{\mathrm{sum}} + \mathcal{L}^{(s)}$

// Final Monte-Carlo estimate (variance decreases with $S$)

**22**    $\mathcal{L}_{\mathrm{clu}} \leftarrow \mathcal{L}_{\mathrm{sum}}/S$

// Return clustering loss

**23**    **return** $\mathcal{L}_{\mathrm{clu}}$

---

## 12 Experiments

**Node initialization.** We initialize node embeddings from pre-trained Node2Vec (Grover & Leskovec, 2016) features learned on the latest graph structure. This is a deliberate choice to make a fair comparison with state of the art model in this domain (Liu et al., 2024). Pretraining provides a strong structural prior that captures local and global neighborhood connectivity before temporal updates begin. Pretrained embeddings are updated using the clustering loss and temporal consistency loss to learn the clusters that considers historical interactions among the nodes. Such initialization stabilizes early training, accelerates convergence, and leads to more semantically meaningful clusters, especially when node attributes are sparse or missing. We choose the number of unique node labels as the number of clusters for evaluation purposes. This choice does not affect training, which remains fully unsupervised; it simply provides a consistent reference for comparing the discovered clusters to ground-truth labels at the current timestamp.

**Coherence Score.** The coherence score Halkidi et al. (2002) measures the average intra-cluster similarity across all clusters, with higher values indicating more compact cluster structure. It is defined in the range $[0, 1]$. For each cluster $c$, let $\mathcal{P}_c$ be all unordered node pairs $(i, j)$ within the cluster. Let $d_{\cos}(i, j)$ denote the cosine distance. We convert distances to similarities $(s(i, j)$ and compute per-cluster coherence as-

$$s(i, j) = 1 - d_{\cos}(i, j). \tag{52}$$

$$\text{Coherence}(c) = \frac{1}{|\mathcal{P}_c|} \sum_{(i,j) \in \mathcal{P}_c} s(i, j). \tag{53}$$

Table 5: Hyperparameter Search Space for TGRAIL Model

| Hyperparameter | Search Space |
|---|---|
| *Training* | |
| Learning Rate | $\log\text{-uniform}(10^{-5}, 10^{-2})$ |
| Batch Size | $\{128, 256, 512, 1024\}$ |
| Optimizer | $\{\text{Adam}, \text{AdamW}, \text{SGD}\}$ |
| *Architecture* | |
| Embedding Size | $\{64, 128, 256\}$ |
| *Temporal/Graph* | |
| Negative Size | $\{5, 10, 20, 50\}$ |
| History Length | $\{3, 5, 7, 10\}$ |
| *Temperature* | |
| Temp Max | $\{5, 10, 15\}$ |
| Temp Min | $\text{uniform}(0.1, 1.0)$ |
| Decay Rate | $\text{uniform}(0.5, 0.95)$ |

**Empirical Evaluation.** Using the metrics from Table 4, TGRAIL demonstrates strong clustering coherence, with scores ranging from 0.8767 to 0.7558 despite rapid growth in the number of active nodes (from 71 to 12,214). Temporal alignment follows the expected trend in a rapidly evolving graph: it is high in early snapshots (e.g., 0.831 and 0.994 for Snapshots 2–3) and gradually decreases as the graph undergoes substantial structural reorganization (down to 0.345 by Snapshot 6). These results confirm that TGRAIL produces clusters that remain both semantically coherent and temporally consistent across evolving graph states.

**Discussion** Our experiments evaluate clustering performance across six temporal graph datasets using four standard metrics (ACC, F1, NMI, ARI). As shown in Tables 2 and 3, our method achieves competitive or state-of-the-art performance, suggesting that joint modeling of temporal dynamics and cluster structure improves representation learning in evolving graphs.

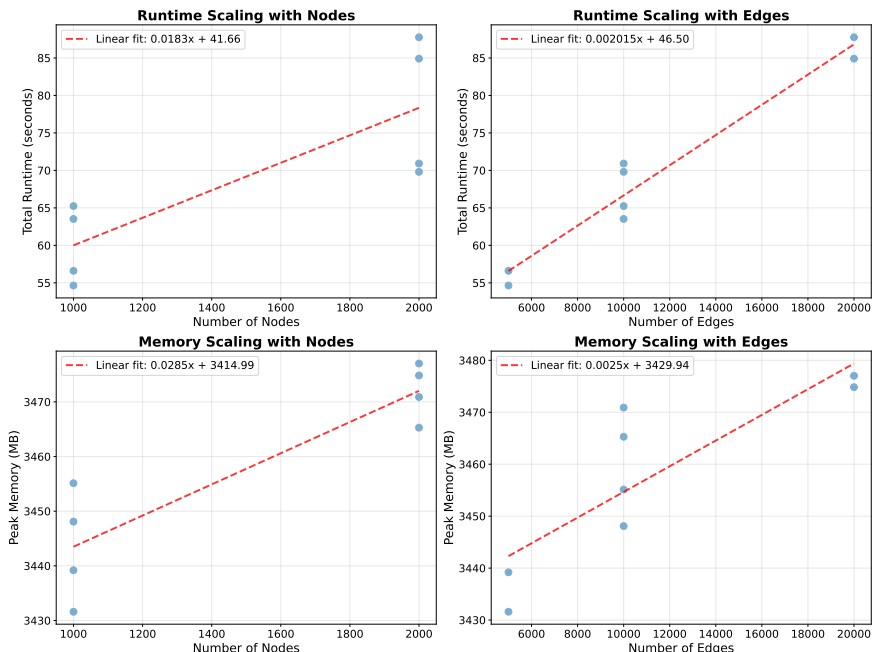

Figure 9: Scalability analysis showing runtime and memory scaling with graph size. Experiments use synthetic graphs with $n \in \{1000, 2000\}$ nodes, $m \in \{5000, 10000, 20000\}$ edges, and $T \in \{10, 50\}$ timestamps. Linear fits (red dashed lines) demonstrate linear scalability.

On the School dataset, where temporal structure aligns cleanly with cluster assignments, our model achieves perfect scores (ACC/F1/NMI/ARI $= \sim 1.00$), demonstrating its ability to recover ground-truth clusters under ideal conditions. In more challenging settings with sparse or noisy temporal signals, such as PATENT and DBLP, our approach outperforms baselines by 3–5% in ACC, highlighting its robustness to incomplete and overlapping event sequences. For datasets with non-stationary dynamics (ARXIV-AI, BRAIN), our model achieves consistent improvements. These results suggest that our temporal encoder captures fine-grained behavioural shifts more effectively than existing methods.

Static baselines (DeepWalk, node2vec, GAE) underperform significantly, reinforcing the necessity of temporal modeling. While TGC incorporates time through Hawkes processes, its decoupled representation and clustering stages limit optimization synergy. In contrast, our fully differentiable framework enables end-to-end learning, aligning temporal representations with clustering objectives.

These findings support our hypothesis that joint optimization of temporal dynamics and cluster assignments improves stability and accuracy in temporal graph clustering. The consistent gains across diverse datasets—ranging from cleanly structured (School) to highly dynamic (ARXIV-AI)—suggest broad applicability to real-world evolving graphs.

**Empirical Complexity Analysis**

To empirically validate linear scalability, we generate synthetic temporal graphs with configurable numbers of nodes, edges, and timestamps, where edges are created using random small scale-free graph structures, timestamps are uniformly distributed across edges, and node features are randomly sampled. As shown in figure 9, our empirical scalability analysis demonstrates strong evidence for linear scaling: runtime scales with edges with $R^2 = 0.952$ and $p < 3.5 \times 10^{-5}$, confirming near-perfect linear scalability, while memory usage scales with nodes with $R^2 = 0.804$ and $p = 0.003$, indicating a very good linear fit that explains 80.4% of the variance. Runtime scaling with nodes shows $R^2 = 0.664$ with $p = 0.014$, representing a moderate linear relationship that is statistically significant, where over 66% of the variance is explained by the linear model. These $R^2$ values, ranging from moderate ($R^2 \geq 0.66$) to excellent ($R^2 \geq 0.80$), combined with statistically significant $p$-values ($p < 0.05$), provide robust empirical justification for our linear scalability claims, as they

Table 6: Runtime comparison in seconds per iteration (mean $\pm$ std) using five different seeds.

| Model | Patent | DBLP | Brain | School | ArxivAI | ArxivCS |
|---|---|---|---|---|---|---|
| | | | Temporal Graph Methods | | | |
| TGRAIL | $2.14 \pm 2.1$ | $12.2 \pm 1.27$ | $95.78 \pm 4.60$ | $9.66 \pm 1.12$ | $36.20 \pm 3.60$ | $59.50 \pm 6.10$ |
| TGC | $2.01 \pm 1.85$ | $11.70 \pm 1.21$ | $94.60 \pm 3.16$ | $9.21 \pm 1.09$ | $34.20 \pm 3.41$ | $57.33 \pm 5.78$ |
| TGN | $4.19 \pm 0.42$ | $24.20 \pm 2.40$ | $196.01 \pm 5.65$ | $19.32 \pm 1.94$ | $65.55 \pm 2.67$ | $95.20 \pm 1.20$ |
| TGAT | $7.39 \pm 1.14$ | $42.30 \pm 2.21$ | $345.22 \pm 3.50$ | $33.20 \pm 3.31$ | $123.10 \pm 4.14$ | $206.10 \pm 6.27$ |
| HTNE | $21.02 \pm 2.30$ | $120.26 \pm 6.21$ | $780.20 \pm 9.80$ | $95.67 \pm 5.53$ | $335.44 \pm 5.56$ | $585.45 \pm 5.90$ |
| Trend | $11.27 \pm 1.33$ | $6.40 \pm 2.64$ | $525.30 \pm 5.30$ | $51.78 \pm 5.11$ | $188.70 \pm 6.67$ | $313.34 \pm 8.65$ |
| | | | Static Graph Clustering Methods | | | |
| Node2Vec | $10.89 \pm 2.33$ | $58.70 \pm 2.60$ | $470.44 \pm 4.70$ | $45.55 \pm 3.45$ | $169.80 \pm 4.70$ | $281.32 \pm 2.80$ |
| GAE | $0.349 \pm 0.30$ | $2.10 \pm 0.87$ | $16.64 \pm 1.69$ | $1.66 \pm 0.37$ | $5.81 \pm 1.88$ | $9.78 \pm 1.71$ |
| SDCN | $0.267 \pm 0.25$ | $1.53 \pm 0.25$ | $1.25 \pm 0.83$ | $1.27 \pm 0.52$ | $4.51 \pm 1.15$ | $7.43 \pm 2.14$ |
| DMoN | $0.271 \pm 0.45$ | $1.58 \pm 0.95$ | $1.27 \pm 0.93$ | $1.25 \pm 0.82$ | $4.51 \pm 1.12$ | $7.58 \pm 1.35$ |

demonstrate that the dominant scaling behaviour is linear with only minor non-linear components that do not substantially impact the overall scalability characteristics of the model.

In Table 6, we show the time requires for each method for each dataset. It is evident that temporal graph methods are more expensive as these methods model graph sequences whereas a static methods consider the graph structure at the final timestamp with no temporal module. Our proposed approach TGRAIL consistently almost faster than two stage clustering techniques such as TGN across datasets. The reason is clear- TGN requires temporal emebedding computation, memory update and perform a post-hoc clustering. Compared to this, TGRAIL performs direct clustering inference in a single model avoiding additional clustering stage after learning the node representations.

