# OpenReview forum: "Differentiable Cluster Discovery in Temporal Graphs"
_TMLR — Accepted by TMLR_

### Review · Reviewer_p4aD · 2026-02-02

**Summary Of Contributions:**

This manuscript considers the graph clustering on (discrete-time) temporal graphs, a challenging yet realistic problem. A new TGRAIL (Temporal Graph Alignment and Index Learning) method was proposed. It was claimed to be a differentiable end-to-end framework with theoretical guarantees for its unbiased and low-variance gradient estimation. TGRAIL formulates the cluster assignment process as stochastic sampling from a Gumbel-Softmax distribution and further enables discrete assignments to be learned through gradient-based optimization.

**Strengths**

**S1**. Graph clustering on temporal graphs is a challenging yet more realistic problem seldom considered in related studies.

**S2**. The proposed TGRAIL is with theoretical guarantees about its unbiased and low-variance gradient estimation.

**Weaknesses** are detailed later.

**Additional Comments:**

N/A

**Audience:**

No

**Audience Explanation:**

Although the temporal graph clustering problem considered in this manuscript is a challenging setting seldom considered in related studies, one significant limitation of this manuscript is that it still adopts the conventional discrete-time dynamic graphs (DTDGs) , as claimed in Section 2.1. In contrast, most related studies on dynamic graphs, though they may consider problems (e.g., dynamic graph representation learning and temporal link prediction) different from that in this manuscript, focus on the more advanced continuous-time dynamic graphs (CTDGs), which can support a more fine-grained description about dynamic graph topology compared with DTDGs.

Moreover, it seems that the proposed TGRAIL method cannot handle the variation of node sets (and also the induced variation of the number of clusters) in dynamic graphs (i.e., still following the conventional transductive setting) while most related approaches can handle this issue via inductive graph inference.

**Broader Impact Concerns:**

There seem no descriptions about broader impact concerns in this manuscript. Although section 5 is with the title 'Conclusion and Research Impact', the last few sentences cannot be considered as the corresponding descriptions about broader impact.

**Claims And Evidence:**

Yes

**Claims Explanation:**

The proposed TGRAIL is with theoretical guarantees about its unbiased and low-variance gradient estimation. The effectiveness of TGRAIL was also preliminarily validated via a series of empirical experiments.

**Requested Changes:**

Some weaknesses that need significant revisions are detailed as follows.

**W1**. Some details about the temporal graph clustering problem considered in this manuscript (i.e., problem statements in Section 2.1) are missing, which need further clarifications.

  The sequence of adjacency matrices ${\bf{A}}^{(t)}$ that describes the temporal graph topology are not mentioned in Eq. (1). As a result, it is unclear how functions $f_\theta$ and $q_\phi$ capture the variation of graph topology.

  A common challenge in related studies on temporal graphs is that the node set may also vary (e.g., new nodes added to and old nodes deleted from the graph at a specific time step) in addition to the variation of edges. This variation of node set can be tackled via the inductive graph inference in SOTA methods (e.g., TGAT, TGN, etc.) However, it seems that this manuscript still considers the conventional transductive setting with a fixed node set.

  Another significant challenge is the possible variation of the number of clusters (e.g., different time steps may have different numbers of clusters). Nevertheless, it remain unclear whether the number of clusters was assumed to be fixed and whether they can be automatically determined by a method or need to be pre-set. It is also unclear whether the proposed method handle these challenges and how by just reading Section 2.1.

***
**W2**. Many details about the proposed methods are not clearly stated in the main paper. As a result, it is still hard to understand how does TGRAIL exactly work.

  In (9), the definition of $L_{\rm{clu}}$ is missing. Hence, it is unclear why 'the clustering loss $L_{\rm{clu}}$ becomes intractable to compute in closed form'.

  More importantly, most descriptions in Section 3 are about the design of training loss. After reading the whole Section 3 and even the methodology part in appendix, it remain unclear how TGRAIL captures the variation of graph topology (e.g., how to derive the temporal embedding via what architectures or algorithms), which is an essential procedure for all the related studies on temporal graphs. The only pseudo-code (i.e., Algorithm 1) does not provide any information for this missing part.

  In Eq. (13), definitions of ${\bf{z}}_i^{(t)}$, ${\bf{z}}_h^{t'}$, and $\delta_{hj} (t-t')$ are not given.


***
**W3**. The time complexity analysis in the last paragraph of Section 3 is more likely for static graphs rather than temporal graphs.

  Concretely, in the derived complexity (i.e., $O(|E|)$), it is unclear what does $|E|$ (maybe the number of edges) mean for a temporal graph (e.g., total number of edges or the number of edges in a specific time step). The $E$ is also not mentioned in the formal problem statement (i.e., Section 2.1).

  For a method operated on temporal graphs, its complexity is usually also associated with the number of time steps. However, the derived complexity does no follow such a property.


***
**W4**. Overall organization of this manuscript and presentation of some figures need further revisions.

  This manuscript puts related work section and summary of dataset statistics (i.e., Table 4), which is not a good presentation for a journal paper. It is suggested to put them in the main paper.

  The font size in Fig. 2, 3, and 7 are too small, which are hard to read.


***
**W5**. References of this manuscript are relatively too old.

  The newest references are two papers published in 2024. It is recommended to add some more references published in recent years, especially those in 2025 and 2026. It is also suggested to ensure that all the references are with the same formate. For instance, in the current manuscript, some venues are with abbreviations but some others are with full names.

***
**W6**. There are no discussions about limitations of this work and possible solutions as future research directions.

****
**W7**. This manuscript does not (anonymously) provide its code to ensure reproducibility of experiments.

---

> ### Author Response · Authors · 2026-02-04
> **Response to reviewer p4aD**
>
> Dear reviewer p4aD,
>
> We thank you for your review and suggestions. Please see below our responses-
>
> We would like to clarify two things-
>
> a) We do not treat the underlying graph as discrete, it is treated as a continuous time dynamic graph and the temporal evolution is captured using Hawkes process which is continuous time temporal point process.
>
> b) TGRAIL is agnostic of the choice of graph encoder. It can work on both trunsductive and inductive setting. Following SOTA models, we chose to use node2vec for fair comparison as the graph encoder, which is an experimental choice not TGRAIL’s limitation.
>
> > Moreover, it seems that the proposed TGRAIL method cannot handle the variation of node sets (and also the induced variation of the number of clusters) in dynamic graphs (i.e., still following the conventional transductive setting) while most related approaches can handle this issue via inductive graph inference.
>
> Inductive graph inference is useful for new nodes representation without retraining the model, which is not the case for clustering objective. In our experiments, we consider temporal graphs with historical interaction patterns where all nodes are available. Inductive methods such as TGAT, TGN handles streaming nodes for future link prediction, however, for clustering, they require post-hoc kmeans to find meaningful communities which separates representation learning objective from clustering objective and results in inferior clustering performance.
>
> We use a fixed number of clusters for consistent evaluation purposes only and unused clusters naturally fades out if there’s no node to assign to that community (for example in Table 3, only four clusters have assigned nodes).
>
> > W1
>
> As mentioned TGN, TGAT etc. method requires a post-hoc clustering training to detect communities, which is not appropriate for optimal clustering and not relevant to our setting. In our setting, we follow a Hawkes self exciting temporal point process where recently interacting nodes increase the likelihood of belonging to the same community, otherwise their likelihood decays following an exponential decay kernel (Equaiton 13).
>
> We have updated equation 1 to clarify how evolving temporal pattern is encoded by function, f. Given the historical interactions of each node upto time $t$, our goal is to learn,
>
> $$
> \mathbf{Z}^{(t)} = f_\theta(\mathbf{X}^{(\leq t)}, \mathbf{A^{(\leq t)}})
> \quad\text{;}\quad
> \boldsymbol{\Pi}^{(t)} = q_\phi(\mathbf{Z}^{(t)}).
> $$
>
> Where
> $\mathbf{Z}^{(t)} \in \mathbb{R}^{|\mathcal{V}| \times d}$: All node embeddings at time $t$
>
>
> As mentioned, we use fixed number of clusters for evaluation consistency similar to SOTA and we discussed this in the limitation section (page  23). In revised manuscript, we will bring it to the main body.
>
> >W2
>
> Complete definition of $\mathcal{L}_{clu}$ is given in equation 11. In equation 9, the intractibility arises from computing the exact expectation term, which requires integrating over the K-dimensional Gumbel distribution ($g \sim \text{Gumbel}(0,1)$ are random variables).
>
> TGRAIL updates the temporal node embeddings following a self exciting temporal Hawkes process. Given a node and its temporal interaction with other nodes, we apply contrastive temporal loss defined in equation 14, where positive samples are the interacting nodes and negatives are non-interacting nodes. This loss maximizes the temporal likelihood of interacting nodes to be in proximity in the latent space. We respectfully refer the reviewer to the following papers:
>
> Yuan, Baichuan, et al. "Multivariate spatiotemporal hawkes processes and network reconstruction." SIAM Journal on Mathematics of Data Science 1.2 (2019): 356-382.
> Zuo, Yuan, et al. "Embedding temporal network via neighborhood formation." Proceedings of the 24th ACM SIGKDD international conference on knowledge discovery & data mining. 2018.
>
> $\bf{z}^{(t_{i})}$ is defined in page 4 before equation 4. We have updated the manuscript $\delta_{hj}$ as learnable decay parameters. Thank you for identifying this.
>
> >W3
>
> For complexity, we have fixed the notation. $O(\mathcal{E})$ is the total number of temporal edges ($O(\sum_{t=1}^{T} \mathcal{E}^{(t)})$) and we will further clarify this in the revised manuscript. Other than this, the complexity discussion section is valid for TGRAIL.
>
> >W4
>
> We sincerely thank you for the comment. We will address this in the revised manuscript.
>
> > W5
>
> Thank you for your suggestion. Our considered baseline models are the latest in this field to the best of our knowledge. We will revise references and include relevant new references with consistent formatting.
>
> > W6
>
> Limitations and future research directions are discussed in page 23, section 12. We will bring it to the main body.
>
> > W7
>
> We have added an anonymized repo here https://anonymous.4open.science/r/tgrail-4B76/README.md
>
> Again thank you for your valuable feedbacks. Please let us know if you have any questions.

---

> ### Comment · Reviewer_p4aD · 2026-03-12
>
> Although the authors have given reposes to other reviewers, there are no point-to-point responses w.r.t. my concerns. As I can check, most of my concerns have not been properly addressed.

---

> > ### Author Response · Authors · 2026-03-13
> >
> > Dear reviewer p4aD,
> >
> > We responded to your feedback first. For some reason it was not showing to everyone. We apologize for this. Please see our responses now that we have updated the readers.

---

> > > ### Author Response · Authors · 2026-03-14
> > >
> > > Dear reviewer p4aD,
> > >
> > > In our revised manuscript, we have added a literature review and dataset characteristics in the main body. Added recent related articles as reference and ensure formatting consistency in the bibliography. Additionally, we have updated the function in equation 1, including the node features and graph structure up to the latest timestamp to learn the latent representation. A separate section for research impact and limitation is also added. We thank you for your feedback and are happy to address any concerns that you may have.

---

### Review · Reviewer_mpnN · 2026-02-18

**Summary Of Contributions:**

The paper is about temporal graph clustering, which is an interesting topic. The authors propose a novel end-to-end framework for temporal graph clustering based on Gumbel–Softmax sampling. The paper is well written and well organized. However, there are several concerns in the current version of the paper that addressing them will increase the quality of this paper.

**Audience:**

Yes

**Audience Explanation:**

The topic the paper focuses on is meaningful, and the authors try to address the deep problems in the field.

**Claims And Evidence:**

Yes

**Claims Explanation:**

The paper is well structured and easy to understand.
The authors conducted experiments to support their ideas.

**Requested Changes:**

1 The paper points out a contradiction between cluster distribution and constantly changing time. Does the paper consider "real-time clustering based on time changes"?

2 Did the author conduct case studies to explore what percentage of nodes change their cluster labels during actual training? Furthermore, what are the behavioral patterns/characteristics of these nodes that change cluster centers?

3 Why is the TGRAIL method better on sparse datasets? What are the characteristics of the distribution of such datasets?

4 In addition to introducing its limitations, the authors could further discuss the potential opportunities for future time series graph clustering and its possible application scenarios.

5 Some important works of temporal graph clustering need to be introduced in the Related Work Section.
[1] Deep Temporal Graph Clustering: A Comprehensive Benchmark and Datasets. IEEE TPAMI.
[2] Multiview Temporal Graph Clustering. IEEE TNNLS.

6 Authors should be careful to check for citation and spelling errors, such as the invalid citation to SDCN on page 15.

---

> ### Author Response · Authors · 2026-03-09
> **Response to reviewer mpnN**
>
> We sincerely thank the reviewer for the positive assessment of our work and for the constructive suggestions that help improve the manuscript. Please see below our responses-
>
> > The paper points out a contradiction between cluster distribution and constantly changing time. Does the paper consider real-time clustering based on time changes?
>
> In our approach, we focus on offline temporal clustering setting where the entire sequence of temporal interactions are available during training. For real-time or streaming clustering, node assignments must be updated online which can be easily incorporated in our framework by applying nearest neighbor search. We have updated this in our future research direction discussion.
>
> > Did the author conduct case studies to explore what percentage of nodes change their cluster labels during actual training? Furthermore, what are the behavioral patterns/characteristics of these nodes that change cluster centers?
>
> Our evaluation focuses on cluster accuracy and temporal consistency. To analyze the dynamics of node cluster transitions, we studied patent dataset and has shown the percentage of nodes that change cluster as new interactions appear in the temporal graph. Intuitively, the change occurs because of new research collaborations in this citation graph.
>
> > Why is the TGRAIL method better on sparse datasets? What are the characteristics of the distribution of such datasets?
>
> In TGRAIL, using gumbel softmax based sampling of cluster assignment in an end to end manner allows us to have a stable cluster distribution for sparse graph where sparsity causes instability in nodes representation. Additionally, sparse graphs have limited structural information, using the historical interactions TGRAIL stabilizes the cluster assignment.
>
> In real world, sparse temporal graphs are common where the average node degree is low compared to the toal number of nodes. In the revised manuscript, we have added Table 1 describing the characteristics of the datasets used for experiment.
>
>
> > In addition to introducing its limitations, the authors could further discuss potential opportunities for future time series graph clustering and its possible application scenarios.
>
> In the revised manuscript, we have added a dedicated section describing the research impact and future work.
>
> > Some important works of temporal graph clustering need to be introduced in the Related Work Section.
>
> We have included this recent works in our revised manuscript. Thank you for pointing this out.

---

> > ### Comment · Reviewer_mpnN · 2026-03-12
> > **Comment**
> >
> > Thanks for the response, I have no questions.

---

### Review · Reviewer_qAYh · 2026-02-24

**Summary Of Contributions:**

The paper claims to present and solve a new task: given a time series of graphs, learn clusters in the graphs at each time point, modelling the evolution of clusters over time. The proposed method (TGRAIL) is end-to-end differentiable and provably consistent. Experiments compare TGRAIL to a selection of other methods over a number of real datasets.

Strengths:
- sensible/natural approach to the task
- consistency proofs
- complexity analysis (and favorable scalability)
- differentiability will enable easier integration with other deep learning methods

Weaknesses:
- No background/related works section
- Questionable metrics used for empirical evaluation
- No concrete application examples
- No mention of opensource software/implementation details

**Audience:**

Yes

**Audience Explanation:**

The paper addresses a natural question (how to learn clustering of graph data that evolves over time) in a relatively general and straightforward manner.

**Claims And Evidence:**

No

**Claims Explanation:**

The mathematics all seems reasonable, but (i) the connection to existing work is unclear, (ii) the empirical results need more details to be interpretable and trustworthy, and (iii) the motivation and claimed use-cases remain unclear.

**Requested Changes:**

Critical (in order of importance):
1. explain how the output of TGRAIL (time-indexed clustered graphs) is compared to the other methods (which all output a static graph?)
1. add an explicit *Related Work* subsection. A quick search turned up some related work ([doi](https://doi.org/10.1063/5.0228419)), and I suspect there's more. The current presentation implies that no other work has addressed this problem before.
1. find some way to quantify uncertainty of the numbers reported in tables 1 and 2 (e.g., run multiple replicates and report standard error or confidence intervals)
1. add something like Table 1 but with runtime per method and dataset

Clarification questions:
1. do the different colors of underline in tables 1 and 2 mean anything?
2. the beginning of Section 4 claims many datasets lack reliable ground truths---does this include the 6 datasets used in the experiments?
3. concretely, when might a researcher be interested in the evolution of graph clusters over time? is it possible to include an actual demonstration of this (e.g., using one of the real datasets in Section 4)?

---

> ### Author Response · Authors · 2026-03-06
> **Response to reviewer qAYh**
>
> Dear reviewer qAYh,
>
> We thank you for your review and feedback. Please see below our responses-
>
> > No background/related works section
>
> We presented the related work section in the Appendix (page 14). In the revised version, we have added this in the main body.
>
> > Questionable metrics used for empirical evaluation
>
> We report accuracy, f1, nmi and ari metrics to evaluate our proposed method which is common in the similar kind of works [1][2].
>
> > No concrete application examples
>
> In the third paragraph of the introduction, we mentioned some applications of temporal graph such as tracking evolving communities, anomaly detection and future link prediction with references. In figure 1, we demonstrated how cluster evolves as new interaction appears in a temporal graph. Finally in Figure 5, we showed how communities in patent dataset evolves over time.
>
> > No mention of opensource software/implementation details
>
> Our anonymized code can be found here https://anonymous.4open.science/r/tgrail-4B76/README.md
>
> > explain how the output of TGRAIL (time-indexed clustered graphs) is compared to the other methods (which all output a static graph?)
>
> For a given node at current timestamp and it's historical interactions with other nodes, TGRAIL captures the continuous temporal evolution following a self exciting Hawkes process and assigns the node a cluster using Gumbel Softmax distribution. Other methods such as DeepWalk, GAE, node2vec considers static graph whereas TGN, TGAT etc. methods consider temporal graph. These approaches requires a post-hoc k-means to learn the cluster assignment distribution. Instead of separating representation learning and clustering, TGRAIL unifies them and learns cluster assignment distribution in an end to end manner.
>
> > find some way to quantify uncertainty of the numbers reported in tables 1 and 2 (e.g., run multiple replicates and report standard error or confidence intervals)
>
> Below, we report the mean and std. deviation of TGRAIL and current SOTA model for different random seeds, TGC [2]. In the final version, we will add this for all models.
>
> | Dataset | Model  | ACC (mean ± std)    | NMI (mean ± std)    | ARI (mean ± std)    | F1 (mean ± std)     |
> |---------|--------|---------------------|----------------------|----------------------|----------------------|
> | PATENT  | TGRAIL | **0.522 ± 0.011**   | **0.377 ± 0.025** | **0.340 ± 0.027**        | **0.404 ± 0.009**        |
> |         | TGC    | 0.476 ± 0.002       | 0.3026 ± 0.0162      | 0.2399 ± 0.0136        | 0.372 ± 0.016        |
> | DBLP    | TGRAIL | **0.506 ± 0.007**     | **0.377 ± 0.005**      | **0.226 ± 0.004**        | **0.506 ± 0.023**        |
> |         | TGC    | 0.484 ± 0.018     | 0.371 ± 0.012      | 0.227 ± 0.012        | 0.445 ± 0.031        |
>
> > add something like Table 1 but with runtime per method and dataset
>
> In the main body, we have discussed the computational complexity of our approach and in the appendix, we discussed the empirical computational complexity using a synthetic temporal graph dataset.
>
> > Broader impact
>
> In the conclusion section, we briefly discussed the impact. In the revised version, we have created a separate section discussing the research impact.
>
> > do the different colors of underline in tables 1 and 2 mean anything?
>
> In both tables, black colour is used for underlining. Underline refers to the second-best model according to our evaluation.
>
> > the beginning of Section 4 claims many datasets lack reliable ground truths---does this include the 6 datasets used in the experiments?
>
> No, our considered six temporal graph datasets have ground truths at the final timestamp which is approapriate for temporal node classification. Following [2], we use it for clustering purpose.
>
> > When would a researcher care .. and demonstatation
>
> A researcher would be interested in our approach when they want to track how a community evolves, clusters for anomaly or fraud detection in a dynamic network. In the exeperiment section, we demonstrated how the communities evolve in patent dataset across six different timestamps. Table 3 shows the quantitative evaluation—percentage of nodes changing community, cluster stability, and temporal consistency as new interactions occur.
>
>
> References
>
> [1] Bo, Deyu, et al. "Structural deep clustering network." Proceedings of the web conference 2020. 2020.
>
> [2] Liu, M., et al. "Deep temporal graph clustering." ICLR, 2024.

---

> > ### Comment · Reviewer_qAYh · 2026-03-09
> >
> > Thanks for the response! This clarifies some things, but a few of my original points remain:
> > 1. Why not include a table comparing runtime of the proposed method compared to other methods? (I understand that the paper already discusses the theoretical complexity, but actual runtime numbers are still valuable and should require minimal coding---is there some good reason not to include these valuable empirical results?)
> > 2. How many replicates are used to produce the newly reported mean and standard deviations?
> > 3. It's still unclear how the metrics are used to compare the output of the proposed method (which is time-indexed) is compared to the outputs of other methods (which are static). The metrics (accuracy, f1, nmi and ari) are standard for static clusters, but I still don't see an explanation of how they're used for the time-indexed clusters.

---

> > > ### Author Response · Authors · 2026-03-13
> > >
> > > Dear reviewer qAYh,
> > >
> > > Thank you for the comments. Please see our responses below.
> > >
> > > 1. Runtime comparison: Like you mentioned, we have discussed theoretical time complexity. In the appendix section, we used synthetic data to validate the claimed time complexity of TGRAIL. In the revised version, we have added the time required to complete one iteration for all the methods and datasets in Table 6 in the appendix. In the initial version, we did not include this, as we thought it might not be fair to compare temporal vs. static graph methods.
> > >
> > > Runtime comparison in seconds per iteration (mean ± std) using five different seeds.
> > >
> > > | Model | Patent | DBLP | Brain | School | ArxivAI | ArxivCS |
> > > |------|------|------|------|------|------|------|
> > > | **Temporal Graph Methods** |||||||
> > > | TGRAIL | 2.14 ± 2.1 | 12.2 ± 1.27 | 95.78 ± 4.60 | 9.66 ± 1.12 | 36.20 ± 3.60 | 59.50 ± 6.10 |
> > > | TGC | 2.01 ± 1.85 | 11.70 ± 1.21 | 94.60 ± 3.16 | 9.21 ± 1.09 | 34.20 ± 3.41 | 57.33 ± 5.78 |
> > > | TGN | 4.19 ± 0.42 | 24.20 ± 2.40 | 196.01 ± 5.65 | 19.32 ± 1.94 | 65.55 ± 2.67 | 95.20 ± 1.20 |
> > > | TGAT | 7.39 ± 1.14 | 42.30 ± 2.21 | 345.22 ± 3.50 | 33.20 ± 3.31 | 123.10 ± 4.14 | 206.10 ± 6.27 |
> > > | HTNE | 21.02 ± 2.30 | 120.26 ± 6.21 | 780.20 ± 9.80 | 95.67 ± 5.53 | 335.44 ± 5.56 | 585.45 ± 5.90 |
> > > | Trend | 11.27 ± 1.33 | 6.40 ± 2.64 | 525.30 ± 5.30 | 51.78 ± 5.11 | 188.70 ± 6.67 | 313.34 ± 8.65 |
> > > | **Static Graph Clustering Methods** |||||||
> > > | Node2Vec | 10.89 ± 2.33 | 58.70 ± 2.60 | 470.44 ± 4.70 | 45.55 ± 3.45 | 169.80 ± 4.70 | 281.32 ± 2.80 |
> > > | GAE | 0.349 ± 0.30 | 2.10 ± 0.87 | 16.64 ± 1.69 | 1.66 ± 0.37 | 5.81 ± 1.88 | 9.78 ± 1.71 |
> > > | SDCN | 0.267 ± 0.25 | 1.53 ± 0.25 | 1.25 ± 0.83 | 1.27 ± 0.52 | 4.51 ± 1.15 | 7.43 ± 2.14 |
> > > | DMoN | 0.271 ± 0.45 | 1.58 ± 0.95 | 1.27 ± 0.93 | 1.25 ± 0.82 | 4.51 ± 1.12 | 7.58 ± 1.35 |
> > >
> > > 2. We have used 5 different random seeds to report the mean and standard deviation.  In the revised version, we have updated Tables 1 and 2 for all the methods.
> > >
> > > 3. For evaluation, we do evaluate only at the final timestamp where we know ground truth labels to compute f1,acc, nmi,ari which is similar to static clusters. For intermediate steps, there's no ground truth available; hence, we use sihouette score and consistency score as shown in Table 3 for the patent dataset.
> > >
> > > Please let us know if you have further concerns. We are happy to address them.

---

### Decision · Action_Editor_1GH1 · 2026-04-01

**Recommendation:** Accept with minor revision

**Additional Comments:**

In the initial reviews, reviewers also raised several concerns and potential weaknesses. One reviewer noted that the current approach is limited to discrete-time dynamic graphs (DTDGs), in contrast to more advanced continuous-time dynamic graphs (CTDGs) used in related studies for finer topological descriptions. Furthermore, TGRAIL appears to be transductive, lacking the capability to handle variations in node sets or cluster numbers, a common characteristic of inductive graph inference methods. Additionally, clarification and improvements were requested for both theoretical and experimental aspects.

We appreciate the authors' efforts to clarify problem settings and detailed aspects, and their revisions to address reviewer concerns. Nevertheless, one reviewer retains a few outstanding concerns that require further attention. Specifically:

1. While the model's current limitations are attributed to experimental choices, it still fails to accommodate variations in node sets within dynamic graphs.

2. The proposed method assumes a fixed number of clusters across all snapshots, which may not align with real-world applications where the cluster count can fluctuate with the dynamic topology.

3. The absence of pseudo-code makes it difficult to understand how TGRAIL captures variations in graph topology (e.g., the derivation of temporal embeddings through specific architectures or algorithms) and the overall training and inference procedures. Consequently, the exact workings of the proposed method remain unclear.

4. The font size in most figures is still too small, hindering readability.

5. The provided link to the anonymous GitHub repository has expired, preventing verification of experimental reproducibility.

The authors are requested to address these points in a subsequent revision before the paper is ready for publication.

**Audience:**

Yes

**Audience Explanation:**

This paper introduces TGRAIL, a novel, differentiable end-to-end framework for temporal graph clustering. TGRAIL employs stochastic sampling from a Gumbel-Softmax distribution for cluster assignments, enabling discrete assignments to be learned through gradient-based optimization with theoretical guarantees on gradient estimation.

The initial reviews recognized the paper's strengths. Reviewers highlighted the meaningfulness and challenge of temporal graph clustering, an under-explored area that better reflects real-world scenarios. The proposed TGRAIL method was praised for its sensible and natural approach, supported by theoretical guarantees for unbiased, low-variance gradient estimation and consistency proofs. Its differentiable end-to-end framework facilitates integration with other deep learning techniques, and complexity analysis indicates favorable scalability, further substantiated by well-designed experiments.

As such, this work would be of interest to individuals in TMLR's audience with research interests in graph-based models.

**Claims And Evidence:**

Yes

**Claims Explanation:**

After clarification and revision, the claims made in the revised paper are now supported by stronger evidence.

---

> ### Author Response · Authors · 2026-05-08
> **Response to Action Editor and Summary of Revisions**
>
> Dear Action Editor,
>
> Thank you very much for your positive recommendation of our paper and for the constructive feedback provided throughout the review process. We are pleased that the reviewers and yourself found the TGRAIL framework to be a sensible, natural, and theoretically grounded approach to the challenging problem of temporal graph clustering.
>
> Please see below our modifications for the final submission:
>
> > While the model's current limitations are attributed to experimental choices, it still fails to accommodate variations in node sets within dynamic graphs.
>
> In the revised version, we have clarified that TGRAIL does support variation in the active node set across snapshots. Our formulation in Section 2.1 defines graph structure at time $t$ as $G^{(t)} = (\mathcal{V}^{(t)}, \mathcal{E}^{(t)})$, where $\mathcal{V}^{(t)}$ is the set of *active* nodes at time $t$, and treats the topology as a chronological stream of temporal events. New nodes therefore enter the model as soon as they participate in an interaction: their embedding is initialized from the structural Node2Vec (an experimental choice) prior and is then refined by the clustering and temporal-consistency losses through the Gumbel–Softmax assignment, exactly like any existing node.
>
> This is also reflected empirically. On the PATENT dataset (Table 4), the active node set grows by more than two orders of magnitude across six snapshots (71 → 182 → 456 → 993 → 2,186 → 12,214), and TGRAIL still maintains coherence scores in the 0.76–0.88 range and positive silhouette scores throughout, while the per-snapshot change rate adapts to bursts of new node arrivals. Similar growth in active nodes is present in DBLP, Arxiv-AI, and Arxiv-CS, which we already report in Tables 1 and 2. What we do not currently do is generalize to entirely *unseen* node identifiers at inference time without any interaction history (true inductive cold-start) or to vary the number of clusters $K$ across snapshots. As per reviewers comments, we have moved both items into the Limitations paragraph (Section "Research Impact and Limitations") and frame them as concrete directions for future work (e.g., parameter-sharing inductive encoders, Bayesian non-parametric or adaptive-$K$ extensions).
>
> > The proposed method assumes a fixed number of clusters across all snapshots, which may not align with real-world applications where the cluster count can fluctuate with the dynamic topology.
>
> We agree that the cluster count can fluctuate in real-world dynamic graphs and we would like to clarify again that the fixed $K$ in our experiments is an *evaluation choice*, not an architectural constraint of TGRAIL. As stated in the Experiments section ("Node initialization"), we set $K$ to the number of unique node labels purely so that ACC, F1, NMI, and ARI can be computed against the provided ground-truth labels at the current timestamp, following the same protocol used by prior temporal graph clustering work. Training itself is fully unsupervised and does not depend on this label count.
>
> The TGRAIL framework itself naturally accommodates a time-varying number of clusters $K^{(t)}$. Empirically, this is also reflected in our PATENT analysis (Table 4), where the number of clusters used at each snapshot adapts to the active graph (4 at timestamp ~1 and 6 at timestamps ~2--6) while coherence and silhouette scores remain stable across the 6 snapshots even as the active node set grows from 71 to 12,214.
>
> For fully data-driven adaptation of $K^{(t)}$ in the unsupervised setting, a natural extension that is compatible with TGRAIL is a Bayesian non-parametric prior (e.g., Dirichlet Process / Chinese Restaurant Process) over centroids, in which new centroids are spawned when sustained low-confidence Gumbel--Softmax assignments accumulate.
> We have made this explicit in the Limitations subsection of the "Research Impact and Limitations" section as a concrete direction for future work.
>
> > The font size in most figures is still too small, hindering readability.
>
> Thank you for pointing this out. In the revised version, we have increased the font size in Figures 1 and 2, and have ensured that all in-figure text is set to at least 12pt across all figures for improved readability.
>
> > The provided link to the anonymous GitHub repository has expired, preventing verification of experimental reproducibility.
>
> We apologize for the inconvenience. We have re-uploaded the anonymized code to a new repository without any modifications of our initial submission, which is available at: https://anonymous.4open.science/r/tgrail-FD9E/README.md.  Once the paper is published and the url will redirect to our original repo.
>
> We again would like to thank you and the reviewers for your time and feedback.
>
> Best regards,
>
> The authors

---

> > ### Comment · Action_Editor_1GH1 · 2026-05-11
> > **Revision for camera ready version**
> >
> > Thank you for revising the paper to address the comments.
> >
> > Please further revise the paper, which includes inserting the author information, and resubmit the camera ready version.
> >
> > Action Editor

---

> > > ### Author Response · Authors · 2026-05-12
> > > **Camera Ready Version**
> > >
> > > Dear Action Editor,
> > >
> > > We have reviewed the paper and added authors information in the revised submission. Additionally, we have added a link to our code repository. We thank you for your time and effort in handling our manuscript and for the valuable feedback provided throughout the review process. We sincerely appreciate the reviewers constructive comments and thoughtful suggestions, which have helped us improve the quality and clarity of the paper.